# A scalable CRISPR/Cas9-based fluorescent reporter assay to study DNA double-strand break repair choice

Paris Roidos [1,6], Stephanie Sungalee [2,5,6], Salvatore Benfatto [1], Özdemirhan Serçin[1], Adrian M. Stütz[2], Amir Abdollahi [3], Jan Mauer [1], Frank T. Zenke [4], Jan O. Korbel [2] & Balca R. Mardin [1✉]

Double-strand breaks (DSBs) are the most toxic type of DNA lesions. Cells repair these lesions using either end protection- or end resection-coupled mechanisms. To study DSB repair choice, we present the Color Assay Tracing-Repair (CAT-R) to simultaneously quantify DSB repair via end protection and end resection pathways. CAT-R introduces DSBs using CRISPR/Cas9 in a tandem fluorescent reporter, whose repair distinguishes small insertions/deletions from large deletions. We demonstrate CAT-R applications in chemical and genetic screens. First, we evaluate 21 compounds currently in clinical trials which target the DNA damage response. Second, we examine how 417 factors involved in DNA damage response influence the choice between end protection and end resection. Finally, we show that impairing nucleotide excision repair favors error-free repair, providing an alternative way for improving CRISPR/Cas9-based knock-ins. CAT-R is a high-throughput, versatile assay to assess DSB repair choice, which facilitates comprehensive studies of DNA repair and drug efficiency testing.

[1] BioMed X Institute (GmbH), Im Neuenheimer Feld 583, Heidelberg 69120, Germany. [2] European Molecular Biology Laboratory, Genome Biology Unit, Heidelberg, Germany. [3] Division of Molecular and Translational Radiation Oncology, National Centre for Tumour Diseases (NCT), Heidelberg University Hospital, Heidelberg, Germany. [4] Translational Innovation Platform Oncology, Merck KGaA, Darmstadt, Germany. [5] Present address: Swiss Institute for Experimental Cancer Research (ISREC), School of Life Science, École Polytechnique Fédérale de Lausanne (EPFL), Lausanne, Switzerland. [6] These authors contributed equally: Paris Roidos, Stephanie Sungalee. ✉email: mardin@bio.mx

DNA double-strand breaks (DSBs) are potentially the most deleterious forms of DNA damage posing a threat to genomic integrity[1]. To repair such toxic lesions and to maintain genome integrity, cells employ two main strategies: end protection and end resection. End protection is carried out by classical non-homologous end joining (c-NHEJ). c-NHEJ repairs DSBs by direct ligation of the broken blunt DNA ends and thus is error-prone[2]. Indeed, lesions repaired by c-NHEJ have a propensity to accumulate point mutations, small insertions, and deletions (InDels). If end protection is not favored, the DSB ends can alternatively be resected, generating 3′ single-stranded DNA (ssDNA) overhangs. Here, the repair can be carried out by three distinct mechanisms: homologous recombination (HR), single-strand annealing (SSA), and alternative end joining (alt-EJ).

HR is a largely error-free mechanism even though the final steps of HR may also require error-prone polymerases[3]. Apart from HR, additional mutagenic end resection-based repair pathways SSA and alt-EJ can contribute to the repair of the resected DSB. SSA occurs between interspersed nucleotide repeats, and although it is based on homology-directed repair, the sequence between the repeats is deleted, resulting in large deletions that can span kilobase-long stretches[4]. Alt-EJ is also initiated by end resection and uses microhomologies of different lengths. Although alt-EJ is reported to be less frequent than c-NHEJ and HR[5], its deficiency can have harmful consequences on genomic integrity since it can lead to chromosomal rearrangements such as translocations[6]. The choice between all these pathways is influenced by several factors, such as the cell-cycle stage or the availability of repair promoting factors. Due to this complex regulation, the outcome of DSB repair may be hard to predict even in controlled settings. However, the way the DSBs are repaired has important consequences not only critical for maintaining genome integrity, but also for cancer therapy. Cancers with DNA repair pathway deficiencies are targeted through ionizing radiation or chemotherapy and, more recently, through advances in the development of specific inhibitors that target several components of the DNA damage repair (DDR) network.

The central DNA damage signaling proteins DNA-dependent protein kinase (DNA-PK$_{cs}$), ataxia telangiectasia mutated (ATM), and ataxia telangiectasia and rad-3-related (ATR) kinases have become promising targets in cancer therapy[7,8]. Furthermore, poly ADP ribose polymerase (PARP) inhibitors have made an accelerated entry into the clinic due to the particular sensitivity of BRCA1-/-2-deficient tumors to PARP inhibition[9–11]. The potential of targeting tumors based on their heightened or altered DNA repair response has led to the development of several compounds that target these DNA repair enzymes. Despite these advances in the generation of DDR inhibitors, simple, yet robust assays to predict and assess the response of cells to these molecules are still missing.

Given the central importance of DNA DSB repair for maintaining genome integrity and in cancer therapy, rapid and precise fluorescent reporters promise to be functional tools to evaluate the DNA DSB repair pathway choices. Over the past decades, several fluorescent reporters have been developed to study specific repair pathways, mainly through the use of the rare-cutting I-SceI endonuclease to induce DSBs resulting in 3′ cohesive ends[12–19]. However, most of these reporters are limited to interrogate one repair pathway at a time, which may not be suitable to characterize the complex DSB repair response. In addition, DSB events that can be induced and tracked by these reporters range from 1 to 25% of the population. Thus, a robust and efficient reporter that allows capturing various responses to DSBs is still missing.

In this study, we develop and utilize a ratiometric fluorescent reporter system, CAT-R, which can simultaneously monitor end protection- and end resection-based DSB repair upon highly efficient clusters of regularly interspaced short palindromic repeats (CRISPR)/Cas9-induced DSBs. This high efficiency and resolution allow us to use CAT-R as a platform to classify small-molecule pharmacological inhibitors and measure the inhibition efficiencies of major classes of DNA DSB repair enzymes. In addition, we combine CAT-R with a custom arrayed genetic screen targeting genes involved in DDR to evaluate the contribution of these genes in DNA DSB repair choice. Furthermore, we present several modifications of the CAT-R reporter that can be easily adapted to different cell-based systems. Taken together, we present a versatile tool that allows easy and simultaneous interrogation of multiple DSB repair pathways that can be coupled to high-throughput screens.

## Results

**Design of the color assay tracing repair system.** The CAT-R reporter consists of two coding sequences for the fluorescent proteins mCherry and eukaryotic green fluorescent protein (eGFP) linked with a self-cleaving P2A peptide. CAT-R leverages the CRISPR/Cas9 system to induce a single site-specific DNA DSB using a guide RNA (gRNA) that targets the eGFP coding sequence. This design allows the mCherry coding sequence to serve as an indicator of end resection events. The repair of this DSB can potentially give rise to three populations with distinct fluorescent signals: (i) small InDels repair-derived frameshift mutations lead to loss of GFP signal ($mCherry^+/GFP^-$); (ii) deletions or other rearrangements larger than ~420 base pairs (bps) lead to loss of both $mCherry$ and $eGFP$ signal ($mCherry^-/GFP^-$) (the mCherry is 420 bp away from the cutting site) and (iii) untransfected cells/error-free repair leaves both $mCherry$ and $eGFP$ sequences intact ($mCherry^+/GFP^+$) (Fig. 1a). We generated and integrated this reporter at a single genomic locus in the human embryonic kidney (HEK)293 and retinal pigment epithelium (RPE-1) cells engineered to express doxycycline-inducible Cas9 (hereafter referred to as HEK293$^{CAT-R}$ and RPE-1$^{CAT-R}$ cells). To establish and optimize the conditions of the DSB induction in CAT-R, we first assessed the reduction of eGFP fluorescence in different CRISPR/gRNA formats and reached the highest efficiency with the use of a synthetic gRNA complex that was then used for all subsequent experiments (Supplementary Fig. 1a). We confirmed the site-specific genome editing events based on an enzymatic mismatch cleavage assay in a time-dependent manner and confirmed the different repair products by microscopy (Supplementary Fig. 1b, c). When comparing the phenotypes observed upon gRNA targeting eGFP with a non-targeting (scrambled) gRNA, as predicted, we observed three populations. We anticipate that the population with intact mCherry and eGFP sequences, is likely a combination of untransfected cells, and in rare cases, DSBs that underwent error-free repair. However, the exact events that lead to the $mCherry^+/GFP^+$ population cannot be resolved with this reporter. For this reason, we focused on the two populations that represent the error-prone repair of the DSB (Fig. 1b; Supplementary Fig. 1d). To confirm that these populations are products of error-prone repair and are stably maintained, we monitored the cells over seven days after DSB induction. We observed that the fluorescence intensity of mCherry and eGFP is reduced over time without drastically changing the ratios among these three populations (Fig. 1c).

In addition, we used flow cytometry to systematically analyze the frequencies of small InDels and large deletions and consistently observed that the two error-prone populations occur at approximately equal frequencies, with the ratio of small InDels to large deletions to be on average 1.18 (±0.23) for HEK293$^{CAT-R}$

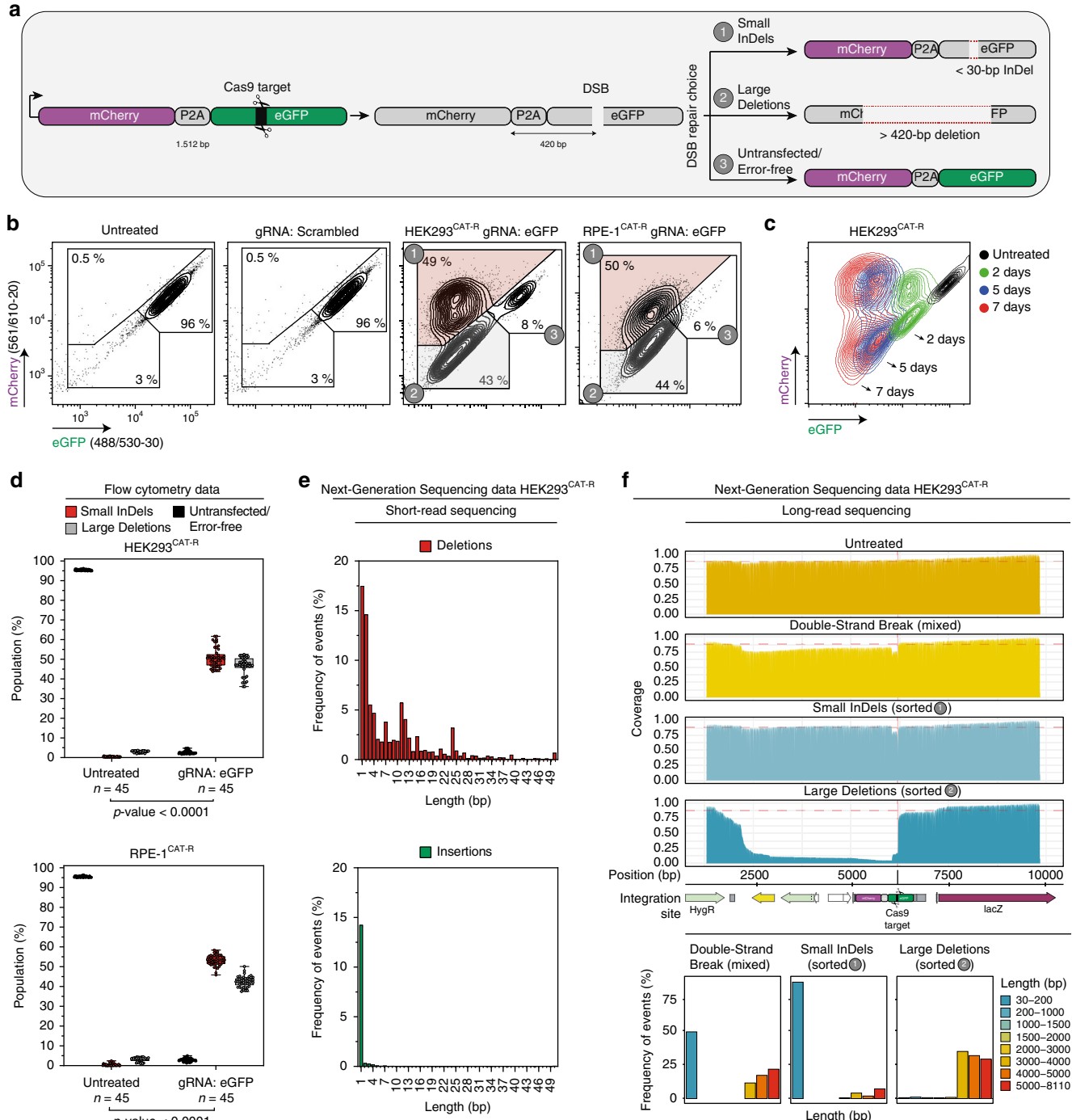

**Fig. 1 The Color Assay Tracing-Repair (CAT-R) reporter system. a** Representation of the CAT-R reporter showing the different DNA repair outcomes after a Cas9-mediated site-specific double-strand break (DSB). The CRISPR/Cas9 target site is indicated at the *eGFP* locus 355 bp downstream of the P2A peptide. If the break resolves through repair with small InDels, frameshift mutations will translate *eGFP* out of frame, and only the *mCherry* will express; if the break resolves with large deletions both *mCherry* and *eGFP* sequences will be lost. For simplicity, we refer to the double negative population as large deletions, but they likely include additional classes of rearrangements. **b** Flow cytometry plots of HEK293[CAT-R] and RPE-1[CAT-R] cells 72 h post-transfection with a non-targeting (scrambled) gRNA as control and a gRNA targeting the eGFP coding sequence. Numbers inside plots indicate percentages of live cells. Axes show fluorescence intensities of eGFP and mCherry proteins. **c** Overlay of flow cytometry plots, monitoring the fluorescent intensity of HEK293[CAT-R] cells for 2, 5, and 7 days following DSB induction. **d** Box and whisker plot ($n = 45$, centerlines mark the medians, box limits indicate the 25th and 75th percentiles, and whiskers extend to min and max, showing all points) of flow cytometry analysis for HEK293[CAT-R] and RPE-1[CAT-R] cell lines 72 h post-transfection with the synthetic gRNA. The *p*-values are calculated with a multiple comparison analysis testing in ANOVA followed by a Dunnett's test. Data are derived from 15 independent experiments; *n* represents the number of all replicates. **e** Short-read PCR amplicon sequencing from genomic DNA harvested 72 h post-transfection to detect small InDels at the targeted site. *Y* axis shows the frequency of events and the *X* axis shows the length of detected deletions or insertions at the break site. **f** Long-read PCR amplicon sequencing from genomic DNA harvested 72 h post-transfection to detect large deletions at the targeted site. *Y* axis shows the coverage of events and the *X* axis shows the length of detected deletions at the break site. Bar plots show the frequency of events per sample.

and 1.24 (±0.20) for RPE-1$^{CAT-R}$ cells. (Fig. 1d) The small difference in the ratio between these two model cell lines may be explained by the slight changes in their cell-cycle profiles, whereby RPE-1 cells spend a longer time in the G1 phase (Supplementary Fig. 1e, f). However, phenotypes observed by CAT-R are independent of the transfection efficiency of the gRNA or the sequence of the gRNA used to target the eGFP, since varying efficiencies of transfection or five independent gRNAs targeting different regions of the eGFP sequence did not heavily affect the ratio between small InDels to large deletions (Supplementary Fig. 1g, h). Therefore, CAT-R represents a robust reporter that can be used to assess the ratio of small InDels to large deletions during DNA DSB repair.

To better understand the repair products in CAT-R, we performed targeted next-generation sequencing (NGS) to detect the composition of InDels generated at the target site of mixed cell population. The maximum length of deletions we detected with short-read sequencing was 171 bp, and for the majority of the cases (98%), the size of InDels was less than 30 bp, counting events with more than 1% frequency. The most common events we observed were 1 bp deletions and 1 bp insertions (Fig. 1e) with 1 bp insertions consisting exclusively of an A at the repair site supporting the idea of templated insertions (Supplementary Fig. 1i), in agreement with predictions of repair products of Cas9-induced DSBs[20–22]. Furthermore, microhomology-mediated end repair events leading to deletions of 11 and 24 bp were also frequent (5.7% and 3.1% frequency, respectively) (Supplementary Fig. 1j). Although, in most cases, the repair of the Cas9-induced DSBs is expected to result in small InDels due to the action of c-NHEJ or alt-EJ, recent studies also suggest that large deletions can occur frequently[23–25]. As these larger events cannot be observed by short-read sequencing, we performed long-read sequencing based on the Oxford Nanopore Technologies (ONT) to detect the composition of large deletions events up to 8.5 kb, generated at the target site in unsorted as well as sorted populations, as indicated in Supplementary Fig. 1k. The median size of the deletions we observed was 4 kb, with a maximum size of 8.1 kb. Interestingly, most of the deletions we observed were larger than 3 kb, with the most common class of deletion events to be between 5000 and 8100 bp having a frequency of 22% (Fig. 1f).

These results suggest that large deletions as a product of end resection events are frequent upon DSB break induction. In addition, we observed, based on the ONT data, that the resection events are asymmetric to the target site, with the majority of the resection events to take place upstream of the protospacer adjacent motif (PAM) site (Supplementary Fig. 1l). These large deletions at least partly may be repaired via microhomology-mediated repair as we observed frequent microhomologies at the break sites, consistent with the current literature suggesting microhomology-mediated repair upon Cas9-induced large deletions[26,27] (Supplementary Fig. 1m). Overall, we demonstrate that based on the color of the populations upon a DSB, CAT-R allows the determination of the frequency of large deletions in addition to small InDels in a quick and robust manner.

**DNA repair deficiencies influence the CAT-R response.** Different pathways compete for the repair of the DSBs. For this reason, we wanted to test if certain DNA repair deficiencies modulate the frequency of the two error-prone populations that can be observed by CAT-R. For this, we first generated single knockout (KO) clones of *PRKDC* and *XRCC4*, (Supplementary Fig. 2a, b) two of the most critical components of the c-NHEJ pathway[28]. In these two cell lines, we evaluated the DSB response with the CAT-R reporter (Fig. 2a). In line with our hypothesis, we observed a substantial reduction in the formation of small InDels

(average: 31% ± 7) in both cell lines upon DSB induction, together with an increased formation of large deletions (Fig. 2b). Next, we tested the effect of end resection factors by targeting critical molecules of HR and Fanconi anemia (FA) pathways such as BRCA1, BRCA2, USP1, FANCF, FANCI, FANCA, FANCD2, FANCE, FANCL, and FANCM. In this case, we transfected cells with synthetic gRNA complexes targeting these genes. Consistent with the idea that end resection can lead to large deletions, targeting these genes led to a decrease in the formation of large deletions concomitant with an increase in the formation of small InDels on average by 10% (±5) (Fig. 2c, d; Supplementary Fig. 2c–e). Altogether, we demonstrate that CAT-R can monitor the shifts in the balance between end protection and end resection-mediated DSB repair.

Having established that CAT-R can respond to defects in end protection and end resection, we next analyzed how the genetic deficiency of the two major DDR enzymes, ATM and PARP1 affects the CAT-R phenotype. ATM phosphorylates a plethora of substrates upon DNA DSBs and is suggested to have critical roles both in end protection and end resection-mediated DSB repair through recruitment of BRCA1 and 53BP1[29,30]. Upon ATM knockout in HEK293$^{CAT-R}$ cells, we observed a decrease in the small InDels formation by 3.2% (±1) and a decrease in the untransfected/error-free repair population by 6.8% (±3.5), concomitant with an increase in the formation of large deletions by 10% (±3.2) (Fig. 2e, f; Supplementary Fig. 2f). These data suggest that resection can still take place even in the absence of ATM, though error-free repair can presumably be negatively affected.

PARP1 has an essential role during the repair of single-strand breaks. However, how it contributes to the repair of DSBs is less well defined[31]. On the one hand, it is suggested to be involved during end resection by rapid recruitment of MRE11 nuclease to the sites of DNA DSBs as well as later stages of HR, presumably by limiting the amount of end resection[32]. On the other hand, PARP inhibition is also reported to stimulate c-NHEJ by interacting with and activating DNA-PK$_{cs}$. To delineate the potential role of PARP1 in resolving DSBs, we generated a *PARP1* KO cell line (Supplementary Fig. 2g) and analyzed the repair. In this case, PARP1 deficiency caused a decrease in small InDels on average by 6% (±1) and an increase in larger deletions, suggesting a more prominent role of PARP1 in either end protection-mediated repair or alt-EJ (Fig. 2e, f), although the effect of PARP1 deficiency was milder than that of c-NHEJ components. Our results indicate that the role of PARP1 in DSB repair is primarily in the first stages of end resection, presumably regulating alt-EJ.

**CAT-R-based screening of clinically relevant DDR inhibitors.** Since the high efficiency of DSB induction in our system allows us to detect even minor changes in DSB repair choice, we next tested whether CAT-R can be utilized as a platform to assess the in vitro potencies of different DDR inhibitor classes. To this end, we selectively targeted DNA repair enzymes currently evaluated as targets in preclinical and clinical trials in a concentration-dependent manner. We initially screened 14 inhibitors (Supplementary Table 1) targeting three members of PI3 Kinase-related protein kinase (PI3KK) family that are involved in DNA damage signaling and repair: DNA-PK, ATM, ATR as well as the CHK1 and Wee1 kinases involved in cell-cycle regulation and analyzed their response by high-throughput flow cytometry (Fig. 3a; Supplementary Table 2).

First, we compared four compounds that inhibit DNA-PK$_{cs}$, the major kinase responsible for cellular c-NHEJ activity[33]. We followed the CAT-R phenotypes upon drug treatment and DSB induction and tested whether any of the inhibitors at any concentration affected the cell viability or the transfection

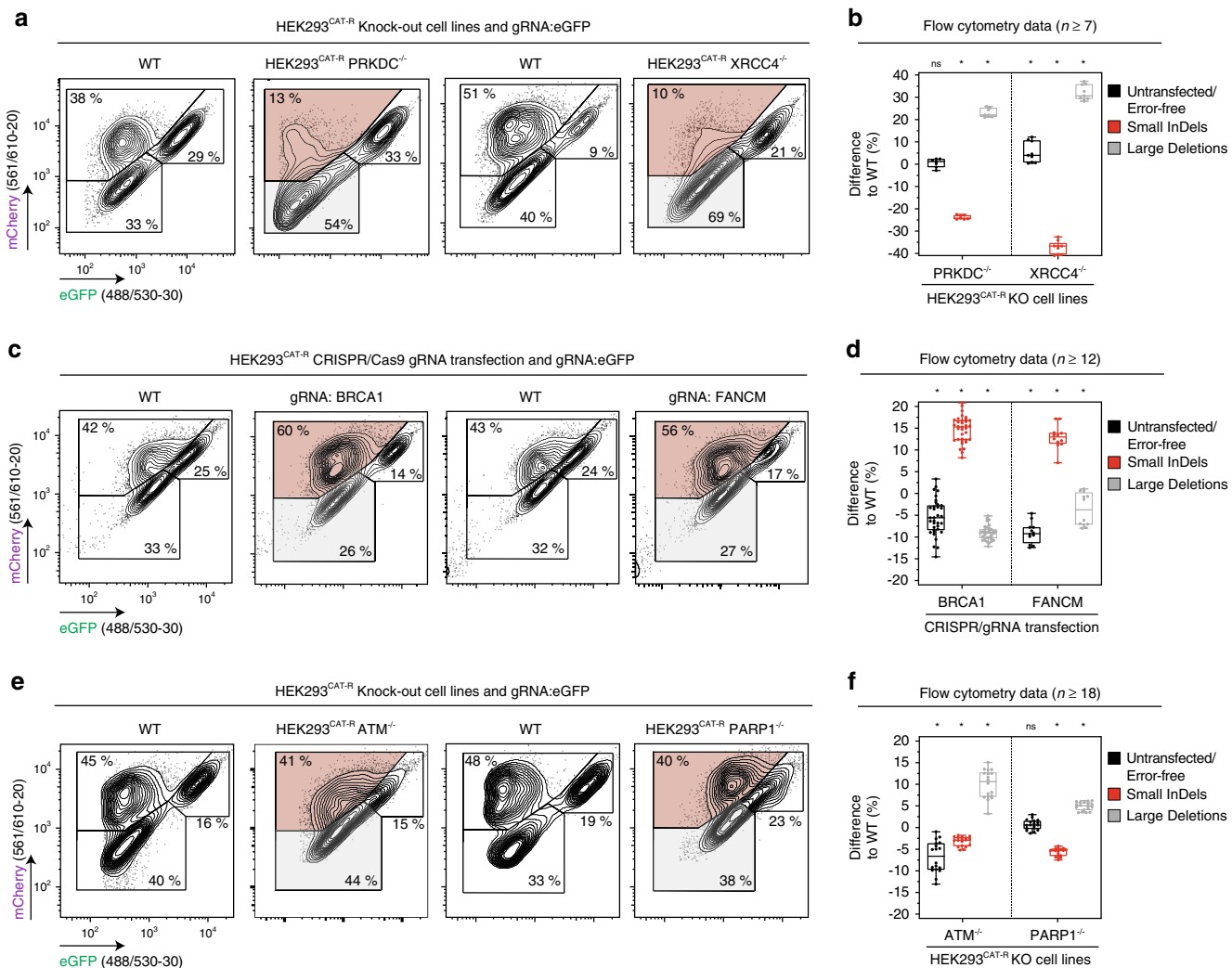

**Fig. 2 DNA repair deficiencies influence CAT-R response.** Representative flow cytometry analysis plots of HEK293[CAT-R] cells 72 h post-transfection with the synthetic gRNA targeting the eGFP coding sequence in **a** *PRKDC* and *XRCC4* KO cells, **c** pool of CRISPR/gRNA transfected cells, and **e** *ATM* and *PARP1* KO cell lines, each compared to their representative WT controls. Numbers inside plots indicate percentages of live cells. Axes show fluorescence intensities of eGFP and mCherry proteins. Box and whisker plots ($n^{PRKDC} = 7$, $n^{XRCC4} = 9$, $n^{BRCA1} = 35$, $n^{FANCM} = 12$, $n^{ATM} = 18$, $n^{PARP1} = 18$, centerlines mark the medians, box limits indicate the 25th and 75th percentiles, and whiskers extend to min and max, showing all points) of flow cytometry analysis for HEK293[CAT-R] deficient or mixed pool CRISPR/gRNA transfected cells are shown in **b**, **d**, and **f**. Values are normalized to wildtype (WT) control, *$p \leq 0.05$ versus WT control, multiple comparison analysis testing in ANOVA followed by a Dunnett's test. All individual *p*-values are included in Source Data file 1. Data are derived from a minimum of three independent experiments; *n* represents the number of all replicates.

efficiency. In agreement with the phenotype we observed with the KO of *PRKDC*, inhibiting DNA-PK$_{cs}$ led to a reduction of small InDels and an increase in large deletions as well as the error-free repair (Fig. 3b; Supplementary Fig. 3a, b). In particular, DNA-PK$_{cs}$ inhibitors KU-0060648 and peposertib (formerly M3814)[34,35] displayed similar pharmacodynamic profiles, showing the relevant phenotype at concentrations as low as at 50 nM by reducing the formation of small InDels on average by 18% (±4) and increasing large deletions by 19% (±4), without a prominent effect on cell viability or transfection (Supplementary Fig. 3c–e). In contrast, we observed no significant effects on DNA repair choice using NU7026 at concentrations lower than 500 nM. However, an improved analog of NU7026, NU7441, performed slightly better by reducing the small InDels formation by 8% (±0.7) and increasing the larger deletions by 5% (±0.5) at concentrations higher than 250 nM. Next, we assessed the effects of ATM inhibition with four different ATM inhibitors. For all the

inhibitors tested, we did not observe any significant effects on cell viability or transfection efficiency (Supplementary Fig. 3f–h). Our data revealed similar profiles of ATM to that of DNA-PK$_{cs}$ inhibition, in that the formation of large deletions was increased by 24% (±2.6) and of small InDel was reduced by 26% (±2.8) at 250 nM (Fig. 3c) with AZD0156 displaying the strongest effects. Our results showcase the utility of CAT-R in detecting minor changes in inhibitor activities.

The CAT-R readout suggested that the inhibition of ATM shows similar phenotypes to that of DNA-PK$_{cs}$ inhibition, and thus may be primarily involved in the formation of small InDels. To exclude potential indirect effects of these compounds, we additionally measured the effect of selected inhibitors on cell-cycle progression and found that only the most potent ATM inhibitor AZD0156[36] slightly increased the percentage of cells in S phase, though this effect does not explain the drastic decrease in the formation of small InDels observed upon ATM inhibition

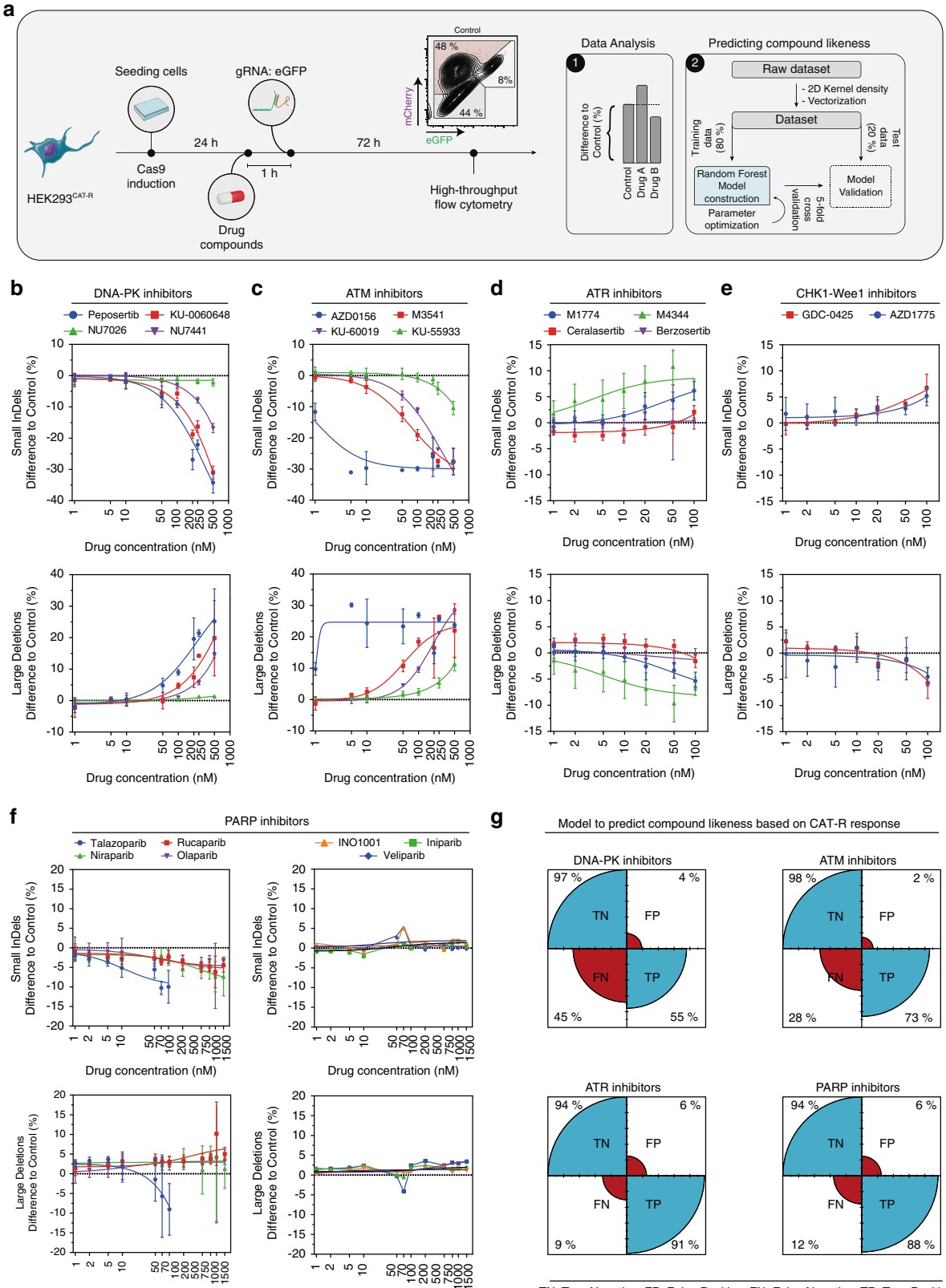

(Supplementary Fig. 3i). In addition, we compared the effect of DNA-PK$_{cs}$ inhibition alone to combined inhibition of ATM and DNA-PK. The effect that we observed with DNA-PK$_{cs}$ inhibitor was exacerbated by combined inhibition of ATM and DNA-PK$_{cs}$, suggesting an additive effect of combined ATM and DNA-PK inhibition (Supplementary Fig. 3j, k). These results together suggest that upon inhibition of ATM there is an increase in large deletions, and a decrease in c-NHEJ/alt-EJ-mediated InDel formation.

In contrast to the effect we observed after inhibiting ATM and DNA-PK$_{cs}$, inhibition of ATR using four different compounds[37] increased the frequency of small InDels on average by 3% (±2)

**Fig. 3 A platform to screen relevant DNA damage repair inhibitors. a** The workflow of the small pharmacological compound screen. HEK293[CAT-R] cells induced with doxycycline (1 μg/ml) were seeded on a 96-well plate, 24 h later the cell culture medium was supplemented with the drug compounds. An hour afterwards, the synthetic gRNA: eGFP was transfected into the cells. Three days post-transfection, cells were analyzed in a high-throughput flow cytometer. Data points were averaged and normalized to the control (DMSO). Line plots ($N^{DNA-PKi} = 6$, $N^{ATMi} = 6$, $N^{ATRi} = 6$, $N^{CHK1-Wee1} = 6$, $N^{PARPi} = 9$, mean ± standard deviation) of flow cytometry analysis for HEK293[CAT-R] cells demonstrate the effect of **b** DNA-PK$_{cs}$, **c** ATM, **d** ATR, **e** CHK1 & Wee1, and **f** PARP inhibitors on small InDels or large deletion formation. Nonlinear regression curves were calculated with the least-squares fitting method using a dose–response model. $N$ represents the number of independent experiments. **g** Fourdotplot for each drug compound class. The total number of analyzed observations for HEK293[CAT-R] cells is 2443. To train the model a repeated cross-validation strategy is used (5-fold, 10 repeats) and an up-sampling technique to balance the compound classes. A confusion matrix is used to evaluate the performance of the test in terms of sensitivity and specificity.

with a concomitant decrease in the frequency of large deletions (Fig. 3d; Supplementary Fig. 4a, b). A pair of highly selective and potent small-molecule inhibitors of ATR, M1774, and M4344 showed the highest efficiencies with CAT-R and reduced the formation of large deletions by 10% (±4.2) and 3% (±1.8), respectively, while increasing the formation of small InDels. However, both compounds showed signs of increased cytotoxicity at concentrations greater than 50 nM and thus were excluded from our analyses (Supplementary Fig. 4c–e). Similar to the ATR inhibitors, inhibition of CHK1 and Wee1 by GDC-0425 and AZD1775, respectively showed similar trends in that they increased the formation of small InDels on average by 8% (±2) at 100 nM concentration without having a major effect on the cell cycle (Fig. 3e; Supplementary Fig. 4f–i). Taken together, we conclude that CAT-R can efficiently monitor the in vitro activities of several inhibitor classes that target the major kinases of the DDR.

**CAT-R-based benchmarking of PARP1 trapping potency.** PARP inhibitors are the first clinically approved drugs that exploit the concept of synthetic lethality. They show promising results in clinical studies as monotherapy for cancers with HR defects[11,38–40]. Clinically used PARP inhibitors differ in their ability to trap PARP1 on the DNA, and trapping potency is crucial for favorable therapy outcomes. To test trapping potency, we compared 7 PARP inhibitors and evaluated their effect on DNA repair choice. In addition to our high-throughput flow cytometry-based assessment of repair choices, we also evaluated the effect of each compound on cell viability (Supplementary Fig. 5a, b). In general, the PARP inhibitors led to a reduction of small InDels and an increase of large deletions on average by 4% (±1.6) at 50 nM (Fig. 3f). These results agree well with the idea that in the presence of PARP, the DSBs are repaired by the alternative EJ pathway, which may contribute to the formation of small InDels due to end protection.

Comparing the potency of all these PARP inhibitors, consistent with the literature, we found that talazoparib had a higher potency compared with the other inhibitors, reducing the frequency of small InDels even at concentrations as low as 10 nM. Increasing concentrations of talazoparib, in particular, higher than 50 nM exhibited a different phenotype, possibly due to the toxic effects of the drug as assessed by cell viability measurements and transfection controls (Supplementary Fig. 5c–f). The next group of inhibitors that are slightly less efficient than talazoparib comprises niraparib, rucaparib, and olaparib. At 200 nM concentration, the average reduction of these three compounds in the formation of small InDels is 4% (1.7) (Fig. 3f). Interestingly, veliparib, iniparib, and INO-1001, a group of compounds that are among the first PARP inhibitors that were later shown not to possess any PARP trapping activity[40], exhibited no prominent effect in the repair of Cas9-induced DSBs (Fig. 3f). These results suggest that CAT-R can also measure differences in PARP trapping activity and can be used as a

screening platform for a rapid in vitro assessment of DDR compound potencies.

To utilize CAT-R for potential drug-screening purposes, we built a machine learning-based model that can predict the class of an unknown compound based only on its CAT-R phenotype. Therefore, we trained a random forest (RF) model with a data set of CAT-R phenotypes (2.443 samples) from known compounds that belong to these four major classes of compounds and built a reference model. The RF model showed an overall accuracy of 83%. Notably, our model showed excellent ability to discriminate the true negatives (high specificity) and accurately predict the classes of the DNA-PK, ATM, ATR, and PARP inhibitor compounds in the test set (Tables 1–2; Fig. 3g). The RF model can be employed to predict classes and efficiency of unknown compounds, based on their similarity with the trained classes that belong to these four major inhibitor compound groups, starting from the output data of the CAT-R reporter.

**A CAT-R-based genetic screen for regulators of DSB repair.** Having established that CAT-R can respond even to minor changes in the DNA DSB repair choices, we set out to investigate the effects of individual DNA repair genes on this process. For this, we designed and synthesized an arrayed gRNA library targeting 417 genes with known and unknown functions in DNA repair. Targeting each gene by two individual gRNAs, we transfected 932 gRNAs, including positive and negative controls[41], and analyzed their effects on the eGFP and mCherry ratios by high-throughput flow cytometry with CAT-R (Fig. 4a; Supplementary Fig. 6a, b). We then calculated Z-scores of all three populations for each gene based on non-targeting (scrambled) controls and formed clusters based on the three populations applying a k-means clustering method. We then performed pathway enrichment analysis on the six clusters that were identified. As expected, c-NHEJ was enriched in the cluster with low Z-scores of small InDels and high Z-scores of large deletions consistent with their phenotype of reduced formation of small InDels. In this cluster, loss-of-function of essential genes for end protection, such as RNF168, TP53BP1, ATM, SETX, and PRKDC, XRCC4 displayed the most considerable differences as compared to the scrambled controls (Fig. 4b, c). On the other hand, loss-of-function of essential genes for end resection, such as the FA components, BRCA1, USP1, COPS4, and BARD1 significantly reduced the formation of large deletions and increased the formation of small InDels. To identify the most active influencers of a Cas9-mediated DSB repair, we additionally used a standard outlier diagnostic tool (Cook's distance) for each gene and found 26 outlier genes (Fig. 4d). Consistent with our clustering approach, we identified several known genes of the c-NHEJ (such as RNF168 and TP53) to be essential for decreasing the rate of small InDel formation and increasing the rate of large deletions along with potentially new regulators of this process such as TTI1 and DDX11 (Fig. 4e, f). TTI1 was previously identified in a genetic screen as a part of a complex that is required for DNA damage signaling to stabilize ATM and ATR[42]. Consistent with these

**Table 1 Machine learning model analysis with Random Forest (model statistics).**

| Class | Sample size | Sensitivity (%) | Specificity (%) | PPV (%) | NPV (%) | Accuracy (%) |
|---|---|---|---|---|---|---|
| Untreated | 507 | 100 | 100 | 100 | 100 | 100 |
| DSB | 170 | 42 | 96 | 50 | 95 | 69 |
| DNA-PKi | 240 | 55 | 96 | 62 | 95 | 76 |
| ATMi | 310 | 73 | 98 | 84 | 96 | 93 |
| ATRi | 452 | 91 | 94 | 76 | 98 | 85 |
| PARPi | 764 | 88 | 94 | 86 | 95 | 91 |

Total sample size = 2443.

*PPV* positive predictive value, *NPV* negative predictive value, *ATMi* ataxia telangiectasia mutated inhibitor, *ATRi* ataxia telangiectasia and rad-3-related inhibitor, *DNA-PKi* DNA-dependent protein kinase inhibitor, *PARP* poly adenosine diphosphate-ribose polymerase, *DSB* double-strand break.

**Table 2 Confusion matrix of the validation data set.**

| Prediction | Sample size | Untreated | DSB | DNA-PKi | ATMi | ATRi | PARPi |
|---|---|---|---|---|---|---|---|
| Untreated | 101 | 101 | 0 | 0 | 0 | 0 | 0 |
| DSB | 28 | 0 | 14 | 3 | 3 | 3 | 5 |
| DNA-PKi | 42 | 0 | 2 | 21 | 10 | 1 | 8 |
| ATMi | 44 | 0 | 0 | 6 | 37 | 0 | 1 |
| ATRi | 82 | 0 | 4 | 7 | 1 | 62 | 8 |
| PARPi | 120 | 0 | 14 | 1 | 0 | 2 | 103 |

Total sample size = 417, accuracy: 83%, 95% CI: (79%, 86%), Kappa: 78%.

*ATM* ataxia telangiectasia mutated, *ATR* ataxia telangiectasia and rad-3-related inhibitor, *DNA-PK* DNA-dependent protein kinase, *PARP* poly adenosine diphosphate-ribose polymerase, *DSB* double-strand break, *CI* Confidence intervals.

observations, our results suggest that TTI1 acts in favor of end protection similar to the loss of ATM. *DDX11* (also known as ChlR1), on the other hand, has an unknown function in the repair of DSBs. It is a DEAH-box DNA helicase that can unwind the DNA with a 5′ to 3′ directionality thus has roles in DNA repair, chromosome structure, and genome integrity[43]. Based on our results, we propose that DDX11 is also a strong influencer of small InDel formation, although the mechanisms of this effect remain to be discovered.

Overall, in our genetic screen, loss of c-NHEJ components decreases the rate of small InDel formation, whereas loss of FA pathway components increases the number of InDels and reduces large deletions (Supplementary Fig. 6c). Interestingly, these analyses also revealed that nucleotide excision repair (NER) components increase the population with the error-free repair while reducing both small InDels and large deletions. NER is an enzymatic pathway that recognizes and repairs a wide range of DNA lesions such as bulky, helix-distorting adducts, or nonhelix-distorting lesions[44]. However, the effects of NER in correcting Cas9-induced DSB have not been studied. Our results suggested that targeting NER components may increase the efficiency of error-free repair in cells upon Cas9-mediated breaks. To test this hypothesis, we modified our reporter to measure single-strand template repair (SSTR). eGFP can be converted to a blue fluorescent protein (BFP) by a single amino acid change, allowing us to measure SSTR by providing a single-stranded oligo-deoxynucleotide (ssODN) template together with the gRNA targeting eGFP[45] (Fig. 5a; Supplementary Fig. 6d–f). To test the effect of crucial NER components in regulating SSTR, we transfected the HEK293$^{CAT-R}$ and the RPE-1$^{CAT-R}$ cells with gRNAs targeting the three independent genes that belong to NER: ERCC3 (XPB), ERCC5 (XPG), and ERCC8 along with PRKDC that was previously shown to increase the rate of knock-ins[46] (Fig. 5b; Supplementary Fig. 6g–i). We then transfected the gRNA targeting eGFP together with ssODN as a template for GFP-BFP conversion. The efficiency of a successful conversion in

HEK293$^{CAT-R}$ and RPE-1$^{CAT-R}$ was on average 4.18% (±1.44) and 7.48% (±2.18), respectively. Although the efficiencies of knock-ins varied in different cell lines, the frequencies of GFP-BFP conversion increased both in the case of genetic depletion of PRKDC or inhibition of DNA-PK$_{cs}$ consistent with the previously reported effects to increase knock-in efficiency by blocking c-NHEJ. When we targeted different components of the NER pathway we also significantly increased the knock-in efficiency up to 12.9% with an average conversion of 5.81% (max fc = 3.88, average fc = 1.82) in HEK293$^{CAT-R}$ cells. Similarly, in RPE-1$^{CAT-R}$ cells deficiency in NER components led to a max increase of 20.8%, with an average of 10.17% GFP-BFP conversion (max fc = 3.64, average fc = 1.71) (Fig. 5b). Together, these results suggest that the NER pathway can be important for mediating SSTR and thus propose an alternative way of increasing the rate of knock-ins in cell lines.

## Discussion

We have developed CAT-R, an in vitro dual fluorescent reporter that can be utilized as a high-throughput tool to interrogate the DSB repair choice. Due to the high efficiency of DSBs introduced by the CRISPR/Cas9 system, CAT-R can simultaneously track the formation of small InDels and large deletions in a ratiometric way with high resolution. We stably introduced CAT-R at a single genomic locus in two non-cancerous cell lines with intact DNA repair pathways. We demonstrated that we could quantify the rates of the populations that underwent error-prone DSB repair while with this reporter, an error-free repair cannot be directly quantified. This is because the error-free population represents a mixture of potential outcomes such as (i) untransfected cells, (ii) cells that underwent homologous recombination, or (iii) some small InDels that are multiplications of three nucleotides, or products of error-free NHEJ[47]. Focusing on the error-prone repair, in both cell lines, we observed similar frequencies in the formation of small InDels

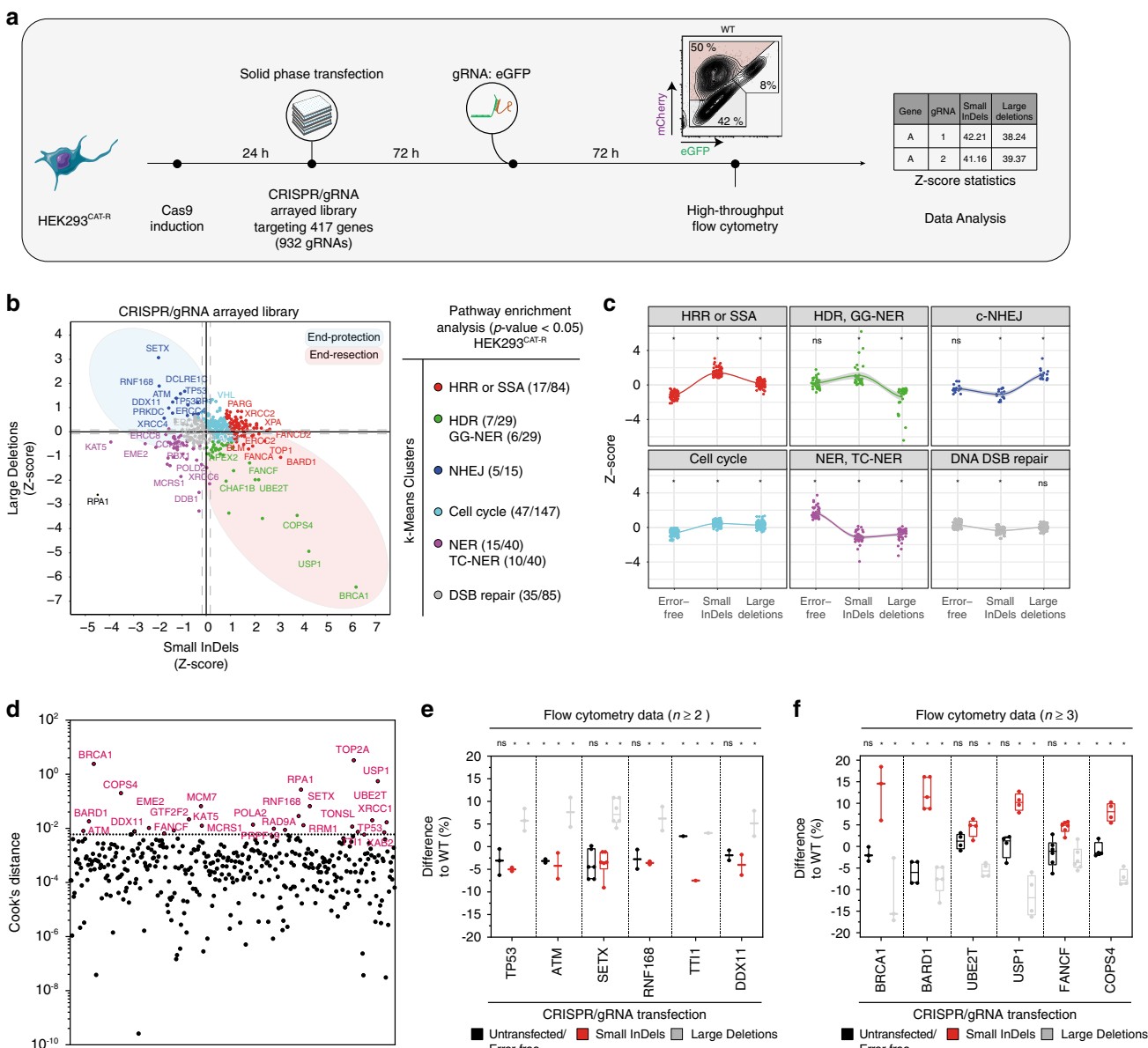

**Fig. 4 The landscape of Cas9-mediated double-strand break repair. a** The workflow of the CRISPR/gRNA genetic screen. In total, 417 genes were targeted with two different gRNA sequences each. HEK293[CAT-R] cells were induced with doxycycline (1 μg/ml), and 24 h later transfected using the solid-phase methodology in pre-coated 96-well plates. 72 h post-transfection, the cells were transfected with the gRNA: eGFP and at day seven analyzed in a high-throughput flow cytometer. Data points were averaged, and the Z-score values were calculated per 96-well plate. **b** A scatter diagram (median) shows the effect of 417 genes upon Cas9-mediated DSB. In the X axis, the regulation of small InDels and in the Y axis, the regulation of the large deletions, are presented. Pathway enrichment analysis of each of the six clusters is shown with the p-value threshold to be set at 0.05. **c** Individual k-means clusters profile in terms of DNA DSB repair choice with the use of the CAT-R system. Each dot represents a gene-phenotype, *p < 0.05 versus WT control, multiple comparison analysis testing in ANOVA followed by a Dunnett's test. All individual p-values are included in Source Data file 3. **d** Cook's distance plot illustrates genes with the most robust phenotype upon Cas9-mediated DSB. The genes passing the significance threshold are annotated. Box and whiskers plots ($n^{TP53} = 3$, $n^{ATM} = 2$, $n^{SETX} = 6$, $n^{RNF168} = 2$, $n^{TTI1} = 2$, $n^{DDX11} = 2$, $n^{BRCA1} = 3$, $n^{BARD1} = 4$, $n^{UBE2T} = 4$, $n^{USP1} = 4$, $n^{FANCF} = 6$, $n^{COPS4} = 4$, centerlines mark the medians, box limits indicate the 25th and 75th percentiles, and whiskers extend to min and max, showing all points) of flow cytometry analysis for the HEK293[CAT-R] cells are shown in **e**, and **f**. Values are normalized to wildtype (WT) control, *p < 0.05 versus WT control, multiple comparison analysis testing in ANOVA followed by a Dunnett's test. All individual p-values are included in Source Data file 3. Data are derived from two independent experiments; n represents the number of all replicates.

and large deletions. Although some of these effects are potentially governed by the cell cycle, our data suggest that the frequency of repair by end resection and the subsequent formation of larger deletions upon Cas9-mediated breaks may not be uncommon. Our CAT-R-based predictions are supported by long-read sequencing and are consistent with recent studies describing more than 20% of larger deletions ranging from 250

bp to 6 kb to occur in mouse embryonic stem cells and RPE-1 cells[21,24]. Based on our ONT data, we also detect a wide range of events that are classified as large deletions. Interestingly, the majority of the resection events are upstream of the Cas9 break site consistent with the earlier reports[48,49] suggesting that upon a DSB, Cas9 remains bound to the DNA leading to asymmetric processing of DNA ends.

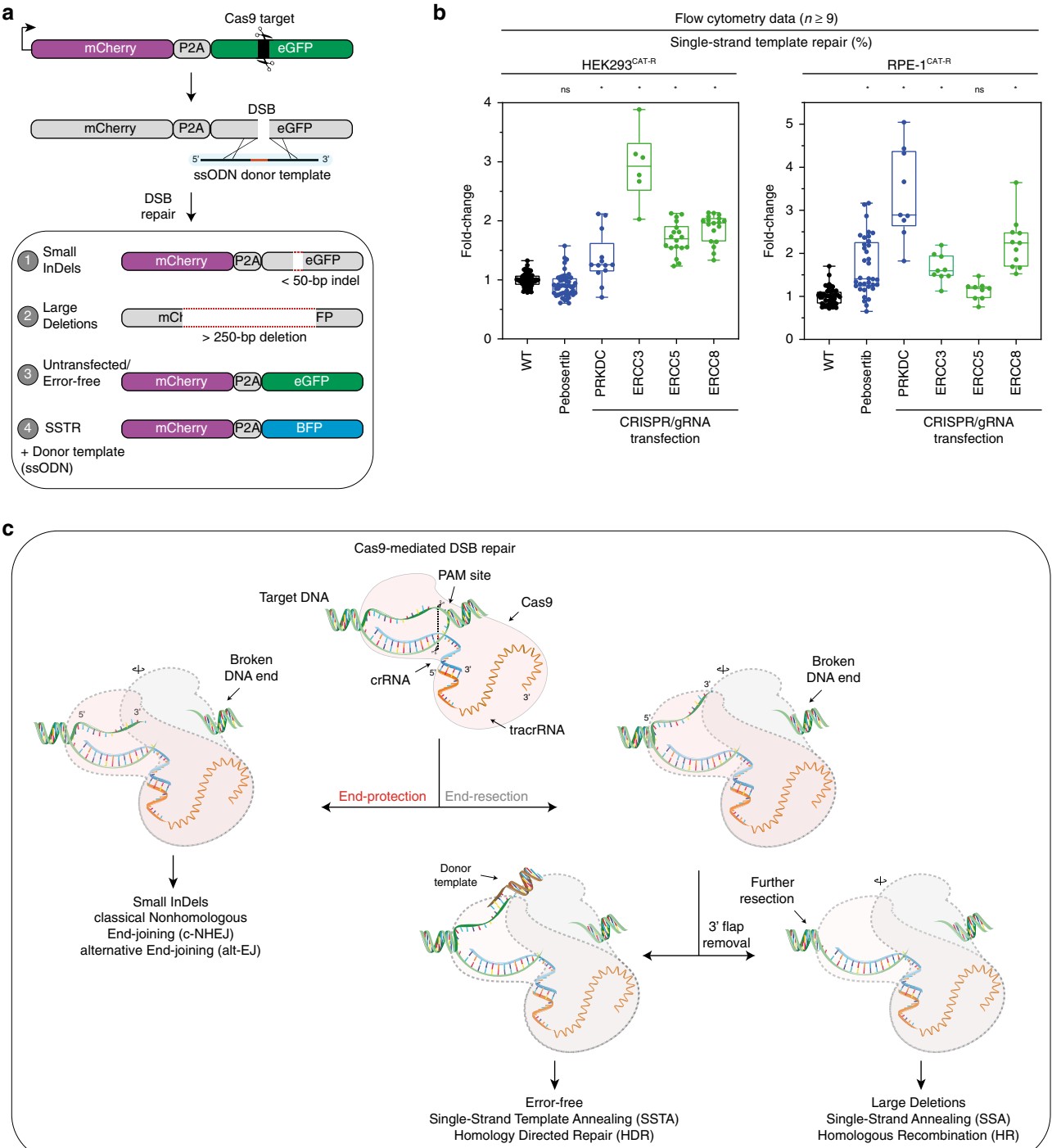

**Fig. 5 Knocking-out NER increases the chances of a successful knock-in. a** Representation of the CAT-R reporter with the use of an external donor template as a ssODN. The ssODN bears the necessary nucleotide changes to convert GFP to a BFP indicating a single-strand template repair (SSTR) event. **b** A box and whiskers plot (For HEK293$^{CAT-R}$: $n^{WT} = 79$, $n^{M3814} = 48$, $n^{PRKDC} = 13$, $n^{ERCC3} = 9$, $n^{ERCC5} = 18$, $n^{ERCC8} = 18$; for RPE-1$^{CAT-R}$: $n^{WT} = 45$, $n^{pebosertib} = 36$, $n^{PRKDC} = 9$, $n^{ERCC3} = 9$, $n^{ERCC5} = 9$, $n^{ERCC8} = 11$, centerlines mark the medians, box limits indicate the 25th and 75th percentiles, and whiskers extend to min and max, showing all points) presenting the frequency of conversion of GFP to BFP with the use of an asymmetric ssODN template in a mixed pool CRISPR/gRNA transfected cells. Values are represented as fold change to wildtype (WT) control, *$p < 0.05$ versus WT control, multiple comparison analysis testing in ANOVA followed by a Dunnett's test. All individual $p$-values are included in Source Data file 3. Data are derived from a minimum of three independent experiments; $n$ represents the number of all replicates. **c** Proposed mechanism of Cas9-mediated DSB repair choice. Once a Cas9-mediated DSB occurs, end protection mechanisms or alt-EJ will act to favor a quick ligation and thus form small InDels. If the damage is not repaired, then end resection mechanisms will start resecting the region leading to large deletions. Due to the nature of Cas9-mediated DSB, a 3′ flap is created, and it is possibly removed with the help of NER components. If this 3′ flap is not removed, then it favors a more efficient knock-in.

While in general the occurrence of the two error-prone populations is approximately balanced, this can be altered by channeling the repair of the DSBs to either end protection or end resection. This occurs because these two major pathways are competing for the repair of the DSBs. Thus, blocking end protection-mediated repair by knocking out its critical components such as PRKDC and XRCC4, increased the frequency of large deletions that are presumably products of end resection. Conversely, when the critical components of resection mechanisms are inhibited such as by the depletion of BRCA1, the rate of small InDels increases probably due to increased availability of end protection proteins to seal the DNA ends[50]. Given that such genetic differences can affect the reporter readout so markedly, one possible future application of CAT-R would be to integrate it into different cancer cell lines with genetic deficiencies in DDR to evaluate how the DSB repair machinery can be affected by different genetic backgrounds of cancer cells.

Inhibiting major components of DDR emerged as a therapeutic strategy for cancer treatment[8,50]. On the one hand, inhibitors against classical DDR kinases such as ATM, ATR, and DNA-PK entered phase I/II clinical trials either as inhibitors for monotherapy or in combination with radio or chemotherapy[51–53]. This strategy is thought to increase vulnerabilities of tumors cells to heightened DNA damage or replication stress[7,54,55]. On the other hand, PARP1 inhibitors that exploit the concept of synthetic lethality have been extensively studied in recent years. However, in many cases, the discovery of additional, more potent, and selective compounds is desirable. For this reason, understanding the profiles of these inhibitors may have important implications for the correct evaluation of their biological effects in DNA repair choice, and their effects in the cells[56–60]. CAT-R enabled us to compare the in vitro drug efficiencies of 21 compounds and to assess the qualitative and quantitative impact of these compounds on DNA repair in terms of small InDel or large deletion formation. We observed particularly pronounced changes in the CAT-R phenotype upon DNA-PK and ATM inhibition and classified compounds consistent with their reported in vitro potency. In addition, we demonstrate that CAT-R can even detect differences in PARP trapping activity, which has been difficult to measure until now and can be used as a screening platform for a rapid in vitro assessment of DDR compound efficiencies. This platform could provide further information on DDR kinase or PARP inhibitor drug discovery, serving as a tool to identify more selective inhibitors. We also developed a machine learning-based strategy to help classify unknown compounds in HEK293[CAT-R] cells, though we note that the model should be reapplied to additional cell lines or different inhibitor classes to adapt to the responses in each cell line and inhibitor class, respectively.

Finally, using CAT-R, we performed a genetic screen, measuring the effect of DDR genes on these three populations. Overall, in our screen, loss of c-NHEJ components decreases the rate of small InDel formation, whereas loss of FA pathway components increases the number of InDels and reduces large deletions. Consistently, FA components have been recently shown to be required for Cas9-mediated SSTR but not for c-NHEJ[61]. In addition to the known components of end protection and end resection, we discovered that knock-outs of components of the NER increase the population that included error-free repair. Since our original CAT-R reporter could not reliably quantify the rates of error-free repair, we further tested this hypothesis by integrating a ssODN-based donor template and measuring the GFP to BFP conversion, thus assessing the rate of SSTR. We showed that in the absence of crucial NER genes, the rate of SSTR events is increased. Although the exact mechanism of how NER can be involved in Cas9-mediated DSB repair remains to be studied in detail, we hypothesize that the

Cas9-induced break may be recognized during transcription, which may then channel the repair to NER. In addition, it has been demonstrated that the gRNA sequence is bound to the antisense strand as an RNA:DNA hybrid and a 5′ to 3′ flap is generated at the non-targeted/sense sequence[62,63], which can be processed by NER. Knocking-out NER may thus increase the chances of a successful knock-in via SSTR/HDR (Fig. 5c).

These results have interesting implications for SSTR-mediated knock-ins since HDR-based genome editing has several potential applications such as the correction of disease-causing mutations. Since in most cases, c-NHEJ is readily available in cells, the majority of DSBs are repaired in an error-prone fashion; thus, strategies to increase the HR-mediated repair are becoming more attractive. So far, inhibition of DNA-PK was shown to increase the rate of HDR by decreasing the accessibility of the c-NHEJ components to the site of repair[46]. Here we provide an alternative approach to increase the rate of knock-ins at transcriptionally active regions. It will be important to see if the effects observed in NER-deficient cells can be applied to other loci and if they can be uncoupled from the cell cycle, thus allowing slow dividing or non-cycling to be edited as well.

In summary, CAT-R can simultaneously measure end protection, end resection, and SSTR-based DSB repair upon a single DSB. Taking advantage of the highly efficient CRISPR/Cas9 system to introduce specific DSBs, we achieve an unprecedented resolution of monitored DSB events, thus can visualize even minor changes in DSB repair activity. The CAT-R reporter can be utilized in several ways and can be adapted in additional model systems to understand how DSB repair choices are made in various cancer cell lines. One alternative approach is via integration of the CAT-R reporter in Cas9-expressing cell lines using the adeno-associated virus site 1 (AAVS1) (Supplementary Fig. 7a). In this efficient system, the CAT-R construct can be inserted into the same chromosomal location as a single stable copy. As proof of principle, we demonstrated the potential use of this approach by integrating this construct in the NCI-H358 lung cancer cell line (Supplementary Fig. 7b). This approach can be used to study how DSB repair choices are made and how they can be influenced in various cancer cell lines. In addition, gRNAs, together with Cas9, can also be stably expressed under an inducible promoter to induce a DSB (Supplementary Fig. 7c). While this system in theory can be advantageous to allow the study of DNA repair in model organisms, we also note that in such systems, unwanted DSBs can be induced due to leakiness of the inducible promoters. Even a few molecules of Cas9 or gRNAs that can go undetected by conventional methods (e.g., immunoblotting) can be sufficient to induce double double-strand breaks, which can be difficult to control (Supplementary Fig. 7d–f), therefore these systems should be used with caution. We anticipate that CAT-R with its versatility can be used as a high-throughput tool and can be easily adapted to chemical and/or genetic screens to assess DSB repair choices.

## Methods
**Cell lines**. Human embryonic kidney (HEK293, Flp-In™ Life technologies), human telomerase reverse transcriptase (hTERT)-immortalized retinal pigment epithelial (hTERT T-Rex™ RPE-1, a kind gift from Jonathon Pines), and the human caucasian bronchioalveolar carcinoma NCI-H358 (ATCC® CRL-5807) mammalian cell lines, were used as model systems. HEK293 cells were cultured in Dulbecco's Modified Eagle Medium, high glucose supplement (DMEM/Gluta-MAX™, ThermoFisher Scientific) containing 10% FBS (ThermoFisher Scientific), 1% Gibco® Antibiotic-Antimycotic. RPE-1 cells were cultured in Dulbecco's Modified Eagle Medium/Nutrient Mixture F12 high glucose supplement (DMEM/F12 GlutaMAX™, ThermoFisher Scientific) containing 10% FBS (ThermoFisher Scientific), 1% Gibco® Antibiotic-Antimycotic. NCI-H358 were cultured in Roswell Park Memorial Institute 1640 Medium (RPMI™ 1640 Media, Fisher Scientific) containing 10% FBS (ThermoFisher Scientific), 1% Gibco® Antibiotic-Antimycotic. All cell lines were cultured at 37 °C and 5% $CO_2$. For the induction of the reporter

and the Cas9 endonuclease, culture media was supplemented with 1 μg/ml Doxycycline for 24 h.

To generate cells that express the reporter as a single stable copy, the FLP recombinase methodology (Flp-In[TM], Invitrogen[TM]) was used for HEK293 and RPE-1 cell lines. More specifically, the Flp-In[TM] T-Rex[TM] system (Invitrogen[TM]) that allows tetracycline-inducible expression was used only for the RPE-1 cell line. For the NCI-H358 cell line, the AAVS1 safe harbor targeting system was used according to the manufacturer's protocol (System Biosciences).

All cell lines contain the reporter at a single genomic locus. In the case of FLP integration, the transfected cells were selected in the presence of 500 μg/μl of neomycin for four days, and a mixed population was generated. For the case of the AAVS1 methodology, the transfected cells were sorted 1–2 weeks post-transfection. After expansion, single-cell suspensions from all cell lines were analyzed, and cells with strong eGFP (488-530/30) and mCherry (561-610/20) signals were sorted using FACSAria I cell sorter (BD Biosciences) to enrich cells harboring the reporter. To generate Cas9 nuclease-expressing cells, the Edit-R[TM] inducible lentiviral particles (Horizon[TM] Dharmacon) were used according to the manufacturer's protocol. The transduced cells were selected in the presence of 1 μg/ml blasticidin for seven days. The expression of Cas9 was controlled by a doxycycline-inducible promoter and the expression was induced with 1 μg/ml doxycycline.

**High-throughput flow cytometry analysis.** Cell populations were gated on a forward- (FSC)/side- scatter (SSC) plot. Cells were further gated on forward-area (FSC-A)/forward-height scatter (FSC-H) plot to determine single cells. Single cells were further gated on side-area scatter (SSC-A)/(405-450/50A) to determine living cells based on DAPI staining. Live cells were further gated to determine eGFP (488-530/30-A)/mCherry (561-610/20-A) cell populations and evaluate in a ratiometric way the fluorescent variations in a FACS LSRFortessa[TM] mounted on high-throughput Samples (HTS) (BD Biosciences, USA). The FlowJo[TM] v10.6.1 software was used for analyzing flow cytometry data.

**Liquid-phase transfection of siRNA.** The eGFP siRNA (Invitrogen[TM]) was used as a positive control for lipofection. Cells were seeded (20,000 HEK293[CAT-R], 8000 RPE-1[CAT-R] cells per 96-well) on 96-well plates (Orange Scientific). After 24 h, cells were 60–80% confluent for transfection. Lipofectamine[TM] RNAiMAX was used as a transfection reagent, and the general instructions for a 96-well plate transfection were followed. The siRNA was combined with Lipofectamine[TM] RNAiMAX in Opti-MEM® Medium to a final concentration of 1 pmol.

**Liquid-phase transfection of synthetic gRNA complexes.** Cells were seeded (20,000 HEK293[CAT-R], 8000 RPE-1[CAT-R], 15,000 NCI-H358[CAT-R] cells per 96-well) on 96-well plates (Orange Scientific), and the culture medium was supplemented with 1 μg/ml doxycycline. After 24 h, cells were 60–80% confluent for transfection. The Alt-R[TM] CRISPR crRNA and tracrRNA (IDT) were used to form the guide RNA complex (gRNA). Each RNA oligo (Alt-R[TM] CRISPR-Cas9 crRNA, tracrRNA) was resuspended in nuclease-free IDTE, pH 7.5 (1× TE solution) to a final concentration of 100 μM. The two RNA oligos were mixed in equimolar concentrations to create a final complex concentration of 3 μM. The gRNA complex was heated at 95 °C for 5 min and then allowed to cool to room temperature (15–25 °C). Lipofectamine[TM] RNAiMAX transfection reagent was used according to the user manual. The gRNA complex was combined with Lipofectamine[TM]RNAiMAX in a ratio of 2:1 in Opti-MEM® Medium to a final concentration of 30 nM.

**Solid-phase transfection of synthetic gRNA complexes.** For experiments using a solid-phase transfection platform, we used flat bottom white 96-well plates (Costar® Assay plate, 3903) and prepared mixtures that are sufficient for 9 wells of a 96-well plate. For each reaction to achieve 2.5 pmol RNA complexes in each coated well, 3 μl Opti-MEM/sucrose solution (1.37% w/v) was mixed with 1.75 μl Lipofectamine[TM] 2000 (Invitrogen, 11668027). To this mix, 6.75 μl of 3.3 μM crRNA: tracrRNA mixture was added, and the final transfection mix was incubated for 20 mins at room temperature. After incubation, 7 μl of gelatin (0.2% w/v in H$_2$O) was added and mixed. The final mixture was diluted in RNA and DNase-free water 1:25 amounting to a total of 450 μl of diluted transfection mixes. From this mix, we plated 50 μl to each well of a 96-well plate. Plates were filled in triplicates and lyophilized using a MiVac vacuum centrifuge, accommodating multi-well plates.

Twenty-four hours before transfection, the culture medium was supplemented with 1 μg/ml Doxycycline. Cells on day of seeding need to be 20–30% confluent and were seeded on a pre-coated flat bottom, 96-well plate (Costar® Assay plate, Corning) (6000 HEK293 and 3000 RPE-1 per 96-well).

**Genomic cleavage detection assay.** Cells were collected 24–72 h after transfection to a 1.5 ml tube. The genomic DNA is isolated according to the manufacturer's protocol using the DNeasy Blood & Tissue Kit (Qiagen). Polymerase chain reaction (PCR) amplification was done, using Q5 Hot Start High-Fidelity 2× Master Mix (#M0494, New England Biolabs) according to the manufacturer's protocol (For oligo design please refer to Supplementary Table 3). The enzyme digest of mispaired dsDNA was done using the Surveyor® Mutation detection kit (IDT)

according to the manufacturer's protocol. The PCR products were analyzed in a 1.5% TBE agarose gel electrophoresis and imaged with a Gel Doc[TM] XR+ (Bio-Rad).

**Western blotting.** Whole-cell lysis extracts of HEK293[CAT-R] and RPE-1[CAT-R] were generated with RIPA buffer (CST—9806S) or custom made HGNT lysis buffer. An equal amount of protein (25 μg/ml) was loaded to a 7.5% precast polyacrylamide gel (Mini-PROTEAN® TGX[TM], Bio-Rad). The cell extracts were transferred to a nitrocellulose membrane (Trans-Blot® Turbo[TM], Bio-Rad) or a PVDF membrane using a transfer apparatus according to the manufacturer's protocols (Bio-Rad). After incubation with 10% nonfat milk in TBS-T (10 mM Tris, pH 8.0, 150 mM NaCl, 0.5% Tween 20) for 30 min, the membrane was washed three times with TBS-T and incubated with antibodies against the protein of interest at 4 °C for 12 h. Membranes were washed three times and incubated with 1:10,000 dilution of IRDye 680RD and IRDye 800CW secondary antibodies for 2 h. Blots were washed with TBS-T three times, developed with the Odyssey system for 2 min (LI-COR Biosciences) and captured with the Image Studio[TM] Lite Software.

The following antibodies were used: Mouse anti-Cas9 (IF, 1:1000, Cell Signaling, cat:14697 s), Rabbit anti-ATM (IF, 1:1000, Cell Signaling, cat:2873 s), Rabbit anti-pChk2 (IF, 1:1000, Cell Signaling, cat:2197 s), Mouse anti-XRCC4 (IF, 1:500, Santa Cruz, cat:sc-271087), Rabbit anti-DNAPK$_{cs}$ (IF, 1:1000, Abcam, cat: ab70230), Rabbit anti-PARP1 (IF, 1:1000, Cell Signaling, cat:9542), Rabbit anti-GAPDH (IF, 1:10,000, Cell Signaling, cat:5174 s), Goat anti-mouse or anti-rabbit IR680 or IR800 (IB, 1:10,000, Licor cat:296-32213 and 296-32211). Uncropped blots can be found in Supplementary Fig. 8.

**Compound screen.** Twenty-five small pharmacological inhibitors were selected to target vital DDR proteins. The compounds were stored in −20 °C as 1 mM stocks in dimethyl sulfoxide. Cells were seeded (20,000 HEK293[CAT-R], 8000 RPE-1[CAT-R] cells per 96-well) on a U-bottom 96-well plate (Orange Scientific), and culture media were supplemented with 1 μg/ml doxycycline. After 1 day, cells were transfected with CRISPR gRNA: eGFP and incubated with the inhibitor compounds for 3 days. Then they were analyzed in a high-throughput FACS LSR Fortessa[TM] analyzer (BD Biosciences).

**Cell viability assay.** CellTiter-Glo® (Promega) was used to determine cell viability, according to the manufacturer's protocol. Cells were seeded (6000 HEK293, 3000 RPE-1 cells per 96-well) on a 96-well white plate with a clear flat bottom (Cost-art® Assay plate, Corning) and cultured for 3 days in the presence of a specific inhibitor. The GloMax®-Multi detection system (Promega) was used as a luminometer to quantify the presence of ATP as an indicator of metabolically active cells.

**Cell-cycle analysis.** Click-iT[TM] EdU Alexa Fluor[TM] 488 Flow Cytometry Assay Kit (ThermoFisher) was used as an assay for analyzing DNA replication in proliferating cells. The Click-iT® EdU protocol was followed according to the manufacturer's instructions. Cells were cultured for 3 days with the presence of inhibitors.

**Drug target validation.** Cells were seeded (300,000 HEK293[CAT-R], 150,000 RPE-1[CAT-R] cells per 6-well) on a 6-well plate. After 1 day, the culture media was supplemented with 4–6 h with the appropriate DNA damaging agent (3 mM Hydroxyurea, Sigma-Aldrich, for ATR inhibitors; 10 μM Bleomycin sulfate, Sigma-Aldrich, for DNA-PK$_{cs}$ and ATM inhibitors) and the specific inhibitor in two concentrations. Afterward, whole-cell lysis extracts of HEK293[CAT-R] were generated with custom made HGNT lysis buffer. The Peggy Sue Simple Western[TM] system was used as capillary electrophoresis to quantify the protein levels of interest according to the manufacturer's protocol (ProteinSimple). A 96-well PCR plate was used to load and prepare the protein sample along with necessary antibodies for Sue assay. The proteins were separated by size, and the normalized signals were evaluated by AUC (area under the curve).

**Colony formation assay.** Exponentially growing cells were harvested and plated in appropriate numbers (250 RPE-1, 1000 RPE-1[TP53, BRCA1−/−] cells per 6-well) on a 6-well plate and cultured (37 °C, 5% CO$_2$) in the presence of a specific inhibitor until cells in control 6-wells have formed sufficiently large clones. After 10–14 days the medium is removed, and the cells were rinsed with PBS. After the removal of PBS, 2–3 ml of 0.15% crystal violet solution was added and left for at least 30 min. The crystal violet mixture was removed, and the 6-well plates were rinsed with PBS and left to dry in at room temperature. For colony quantification, the ImageJ "ColonyArea" plugin was used[64]. Excel was used to analyze the data and generate plots.

**RNA Extraction, cDNA Synthesis, and qPCR.** Total RNA isolation was performed from 10[6] cells using the RNeasy Plus Mini kit (Qiagen) as described in the manufacturer's protocol. The RNA samples were diluted to 250 ng/μl final concentration. All RNA samples within an experiment were reverse transcribed at the same time with the qScript[TM] cDNA SuperMix (Quanta Biosciences) using 500 ng

of RNA as a template and stored in aliquots at −80 °C. Real-time PCR with Fast SYBR® Green (ThermoFisher) detection was performed using a QuantStudio™ 5 Real-Time PCR system (Applied Biosystems™) (for oligo design please refer to Supplementary Table 3). The relative quantification of each sample was performed using the comparative Ct method. The acidic ribosomal phosphoprotein P0 gene (36B4) is used as a housekeeping gene. To compare the transcript levels between different samples the $2 - \Delta Ct$ method was used.

**PCR amplification and Illumina sequencing for detection of small InDels.** For the short-read genome-sequencing assay, DNA was extracted from HEK293[CAT-R] cells using the DNeasy Blood & Tissue Kits (Qiagen) as described in the manufacturer's protocol. After quantification (Qubit™ fluorometer, ThermoFisher Scientific) we employed a two-step PCR protocol (For oligo design please refer to Supplementary Table 3). As suggested in the Illumina protocol for 16S Metagenomic Sequencing Library Preparation, the first PCR step is performed to amplify the targeted DNA region. For each sample, 1 μg of DNA was used to prepare the initial 388 bps PCR amplicon. The 50 μl PCR reactions were set up with the NEBNext® Q5® Hot Start Master Mix (New England BioLabs) and the thermo-cycling conditions were 98 °C for 3 min, 12 cycles of 98 °C for 10 s, 65 °C for 30 s, and 72 °C for 20 s with a final extension at 72 °C for 3 min. To verify the success of the PCR, amplification products were electrophoresed on a 2% agarose gel. The second PCR step was performed to multiplex individual specimens on the same Illumina MiSeq flowcell and to add necessary Illumina adapters. In this second step, primer pairs used contained the appropriate Illumina adapter allowing amplicons to bind to the flow cell, an 8-nt index sequence, and the Illumina sequencing primer sequence. Amplicons were sequenced with 250 bp paired-end reads.

**Data analysis of short-read Illumina sequencing.** The quality control of the reads was performed with FastQC and MultiQC tools. BBMap (v. 38.34) was used for the alignment due to its accuracy to align reads with long InDels. As a reference, the targeted eGFP sequence was used. All the downstream analyses were performed with custom scripts in R (v. 3.4.4). InDels were considered only if they occurred within 1 nucleotide of the Cas9 cleavage site. To guarantee the robustness of the frequency estimation, only events (InDels with a unique position and length) supported by at least 10 reads were considered.

**PCR amplification and long-read sequencing for detection of large deletions.** For the long-read genome-sequencing assay, DNA was extracted from HEK293-[CAT-R] cells using the DNeasy Blood & Tissue Kits (Qiagen) as described in the manufacturer's protocol. After quantification (Qubit™ fluorometer, ThermoFisher Scientific) we performed a PCR step to amplify an 8.5 kb product with a high-fidelity polymerase (PrimeSTAR® GXL, Takara) that generated products with blunt ends. The thermo-cycling conditions were 30 cycles of 98 °C for 10 s, 60 °C for 15 s, and 68 °C for 9 min. For evaluating the sample quality control, we used the Bioanalyzer DNA analysis kit (Agilent Technologies). We followed the suggested protocol (SQK-LSK109) for PCR-free ligation sequencing from Oxford Nanopore Technology (ONT). For every sample, a starting material of 200 fmol amplicon DNA was used for initial end-prep, followed by native barcode ligation with sequential steps of DNA clean-up with magnetic beads (AMPure XP beads, Beckman). Equimolar amounts of each barcoded sample were pooled together to produce a pooled sample of 150 fmol. Following adapter ligation and the DNA clean-up step, the final library was loaded to the MinION flow cell after priming and loading the SpotON flow cell (for oligo design please refer to Supplementary Table 3).

**Data analysis of long-read sequencing.** The base calling and the demultiplexing were performed with guppy_basecaller (v. 3.4.5). The reads with a PHRED quality score lower than 7 were filtered out using NanoFilt (v. 2.6.0). High-quality reads were aligned to the reference sequence with NGMLR (v. 0.2.7) and the structural variations were identified by the tool Sniffles (v. 1.0.11) with a maximum distance to group structural variations together of 2. pycoQC (v. 2.5.0.20) was employed to generate the statistics for each aligned sample. For the SAM to BAM format conversion, the BAM files sorting and indexing and the coverage calculation Samtools was used (v. 1.9). All the downstream analyses were performed with custom scripts in R (v. 3.4.4).

**Genetic background analysis and statistical testing.** We note that we define biological replicates as completely independent experiments carried out on different days with a different batch of materials. In each biological replicate, we always included three replicates that represent, e.g., 3 wells of a 96-well plate. Therefore, N represents the number of biological replicates and n represents the number of all replicates. Results from the reporter are presented as box and whiskers plots with centerlines to mark the medians and the box limits to indicate the 25th and 75th percentiles. Whiskers extend to min and max, showing all points. Data are normalized to gRNA:eGFP WT control. Specifically, for examining the robustness of the method, 151 biological experiments were performed with HEK293[CAT-R] and 56 biological experiments with RPE-1[CAT-R]. The estimates of

significance were determined using a mixed-effect model (Two-Way ANOVA) analysis for multiple comparisons. Every sample is compared to the WT control with the mean of each CAT-R population to be compared with the respective control mean (Source Data 1). Each p-value is adjusted to account for multiple comparisons using statistical hypothesis testing with the Dunnett test. Random effects with zero s.d. were excluded from the model. For data analysis and visualization, the GraphPad Prism 8 was used.

**Compound analysis and statistical testing.** Results from the reporter were presented as the mean ± standard deviation of independent experiments. Each independent experiment entails three technical replicates, N represents the number of biological replicates (Source Data 2). Line plots are used with mean value to be annotated and data to be normalized to gRNA:eGFP DMSO control. The nonlinear regression curve was calculated with the least-squares fitting method using a dose–response model using GraphPad Prism 8 (Supplementary Table 2).

**Random Forest model development.** The raw FACS data were transformed into a vector before being used as input for the Random Forest (RF) model development. Initially, for each sample, the 2D kernel density was computed from the raw FACS data, and the resulted plot was converted into a 100 ×100 pixels image. The image was then flattened into a 10,000 elements vector.

To avoid false predictions, we only included samples where the effect of the compounds was significantly different from the DSB controls (in the absence of any drugs). To identify such cases, we computed the statistical distances of each sample from the controls per 96-well plate using the Kolmogorov–Smirnov test, and we calculated the average. As a reference, we used the distribution of the average statistical distances within the controls; its 95% confidence interval (CI) upper endpoint was used as a cut off to remove samples with a similar phenotype to the control.

Then the data set was split randomly into training and test sets using an 80/20 ratio, preserving the overall class distribution of the data. The training set was used to train an RF model. The hyperparameter tuning was performed randomly, partitioning the data in five equal-sized subsamples of which one was retained as a validation set and the others as a train set. The process was repeated five times so that each subsample was used as a validation set (5-fold cross-validation), then the model showing the highest accuracy was selected. Inside this process, the minority drug class was randomly sampled to be the same size as the majority drug class. To assess the final model, we predicted the classes of the test set and generated the confusion matrix to calculate the performance of the model. The modeling was performed in R using the caret and the ranger packages.

**CRISPR/Cas9 gRNA library.** An arrayed gRNA library was synthesized on 96-well plates (10 × 96-well plates in total) targeting a total of 417 genes (IDT). For each gene, two individual gRNAs were used. On each plate, we used four positive (POLR2A), and six negative (Scrambled, non-targeting gRNA) controls to evaluate the solid-phase transfection efficiency. On the first day, the culture media was supplemented with 1 μg/ml doxycycline to induce the Cas9 expression. On day 2, cells were 60–80% confluent and actively dividing. They were seeded in pre-coated plates containing the gRNA library complexes. Three days post-transfection, the gRNA:eGFP is transfected, and 3 days later, the eGFP and mCherry ratios were assessed by high-throughput flow cytometry.

**CRISPR/Cas9 gRNA library analysis.** Previously, we showed that targeting POLR2A can serve as a positive control of transfection efficiency[41]. POLR2A is an essential gene for the survival of a cell. Therefore, plates in which POLR2A gRNA transfected wells contain more than 1500 cells (indicating lower than 85% transfection efficiency) were removed from the subsequent analysis due to poor transfection conditions. The data from the two replicates were normalized to gRNA: eGFP WT control and then averaged per gene. We also removed 8 genes whose KO resulted in a marked decrease in viability after 5 days from the subsequent analyses. For the remaining 409 genes after this initial filtering step, we calculated Z-scores of all three populations for each gene based on non-targeting (scrambled) controls (Source Data 3). We formed clusters based on the three populations applying a k-means clustering method, then performed a pathway enrichment analysis using Reactome Pathway Database (https://reactome.org) to identify pathways enriched in the given gene list sets. To identify the most influential genes of a Cas9-mediated DSB, we used a standard outlier diagnostic tool (Cook's distance). The significance threshold is calculated with the formula $4/(N\text{-}k\text{-}1)$, where N is the number of observations and k the number of explanatory variables. The analysis was also conducted in R (http://www.R-project.org/) and figures were produced using the package ggplot2.

**Reporting summary.** Further information on research design is available in the Nature Research Reporting Summary linked to this article.

## Data availability
Raw amplicon sequencing data are available at the European Nucleotide Archive with the study number: PRJEB35246 (https://www.ebi.ac.uk/ena/browser/search). All data are

available from the authors upon reasonable request. Source data are provided with this paper.

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

## Acknowledgements

We thank Malte Paulsen and Diana Ordonez from EMBL flow cytometry core facility for assistance with high-throughput flow cytometry, and Anja Telzerow and Vladimir Benes from EMBL GeneCore for their assistance with sequencing. We thank Heike Dahmen and Matthias Heil from Merck KGaA, Darmstadt, Germany for assistance with the automated western blot system. We would also like to thank Anton Khmelinskii for critical comments on the manuscript and John Lindner and Alexandra Vitor for useful suggestions and discussion of experimental procedures that improved this study. J.O.K. and S.S. were funded by the European Research Council (ERC) starting grant, #336045. This work was supported by Merck KGaA, Darmstadt, Germany. Editorial support was provided by David Griffiths, Ph.D., CMPP of Bioscript Group Ltd (Macclesfield, United Kingdom), funded by Merck KGaA, Darmstadt, Germany. Figures 3, 4 and Supplementary Fig. 7 were created using images from Servier Medical Art (http://smart.servier.com). Servier Medical Art by Servier is licensed under a Creative Commons Attribution 3.0 Unported License.

## Author contributions

B.R.M. conceived and designed the study. S.S. and J.O.K. designed the CAT-R system and S.S. generated the HEK293$^{CAT-R}$ cell line. A.M.S. generated the RPE$^{TP53−/−BRCA1−/−}$ cell line. P.R. carried out all the experiments with help from Ö.S. S.B. developed the computational model for predicting drug responses and analyzed the short-read and long-read sequencing data. J.M. designed the SSTR reporter with the ssODNs. A.A. and F.T.Z. provided critical input and materials. B.R.M. and P.R. wrote the paper with input from all authors.

## Competing interests

Frank T. Zenke is an employee of Merck KGaA, Darmstadt, Germany. Paris Roidos, Salvatore Benfatto, Özdemirhan Serçin, Jan Mauer, and Balca R. Mardin are employees of BioMed X Institute (GmbH), Heidelberg, Germany. Amir Abdollahi is an academic mentor at BioMedX and receives research grants from Merck KGaA. The remaining authors declare no competing interests.
