## [Peer Review File · Nature Communications]

Reviewers' comments:

Reviewer #1 (Remarks to the Author):

The manuscript from Dr. Mardin's group describes a colorimetric assay system (color assay tracing repair (CAT-R)) to simultaneously monitor outcomes of DSB repair via end-protection and end-resection pathways. This method relies on CRISPR/Cas9 to introduce DSBs in a tandem fluorescent reporter, which can distinguish small insertions/deletions from large deletions. Further, they validated the utility of the CAT-R system using 1) clinically relevant small molecule pharmacological inhibitors and 2) in a custom-designed genetic screen targeting gene involved in DDR to evaluate the contribution of these genes in DNA DSB repair choice. Notably, they found through the assay system that impairing nucleotide excision repair (NER) favor error-free repair, providing a possible way to increase the rate of CRISPR-based knock-ins by homology-directed repair.

The concept of measuring DSB repair choices in cells has been widely studied in DNA damage repair field, and many similar assay system has been developed Certo, M. T. et al. *Nat. Methods* 8, 671–676 (2011), Gunn, A. & Stark, J. M. *Methods Mol. Biol.* 920, 379–91 (2012), Kuhar, R. et al., *Nucleic Acids Res.* 42, e4 (2016), Moynahan et. Al. *Molecular Cell*, 263-272 (2001). The unique feature of the assay system described in this manuscript is the usage of a CRISPR/ Cas9 to introduce specific lesions in the eGFP gene. They generated a HEK293CAT-R cell-line and tested the system by targeting individual DSB repair genes or chemical inhibitors. Most experiments are well designed, and data are organized in a way to support the main conclusion. However, there are some concerns regarding the novelty of the study and result interpretations. The most interesting observation that the Authors just touched upon was the importance of modulating NER to increase CRISPR-based knock-in efficiency. Developing a story based on that observation would have been more exciting and will be of general interest.

Comments:

- 1) The Authors need to clearly mention superiority of the proposed assay system from other DSB repair assays already present. Additionally, multiple cell lines should be tested as clearly HEK293 and RPE1 showed differences in basal level.
- 2) The double-positive (mCherry and eGFP) population shows both un-transfected and error-free repair active cells. This might affect the interpretation of the overall repair capacity and repair choice of the cells. The author needs to state how they have overcome this issue clearly.
- 3) Between the large deletions (double negative population) Vs. Indels (eGFP negative) there will be a range of other events, is the assay sensitive enough to measure the array of fluorescence change? This might be helpful in understanding the extent of DDR protein's influence on pathway choice.
- 4) Using a CRISPR library to assess the role of different repair proteins on DSB repair choices is a robust way to test the system and identify new dependencies on pathway choice. However, the authors need to address the transfection efficiency of both gRNA to eGFP and CRISPR library to cells. From that screen, authors found that the NER pathway plays an active role in CRISPR-knock in ability (although clear data is missing), this data has a potential and should be expanded and studied with more positive and negative controls to make a claim.

Reviewer #2 (Remarks to the Author):

The manuscript by Roidos et al describes a scalable system to analyze events that can happen at the site of a double-strand break. The system is based on generating a cell line stably expressing (1) two proteins with different fluorescent properties from the same promoter and (2) inducible Cas9 endonuclease. The authors then claim that transfection of a gRNA targeting one of the two fluorescent proteins (mCherry and GFP) would lead to one of three events after induction of Cas9: (1) error-free repair and expression of both proteins, (2) "small" deletions or insertions leading to

loss of that protein or (3) “large” deletions leading to loss of both proteins. The three populations of cells are then detected by flow cytometry. The authors provide evidence that the balance of the three outcomes in this two-color assay could distinguish the choice of the DNA double-strand repair system used to repair the lesion. They go on to show potential applicability of this assay in drug and genetic screen to search for chemical or genetic entities that could affect certain DNA repair pathways by measuring shifts in the three populations of cells.

The method is interesting and potentially applicable for high-throughput screens, however, the manuscript has some major issues related to interpretation, validation and reproducibility of the method that need to be addressed before it can be suitable for publication as described below.

1. The approach has one fundamental flaw – the DNA molecules digested by Cas9 and repaired error-free can also be cleaved again until they are repaired with errors such that they can no longer be substrates for the Cas9 enzyme. Therefore, this is not a very good system to estimate the fraction of error-free repair.
2. Related to the point above, it is impossible to distinguish error-free repair and uncleaved DNA, further arguing that error-free repair can not be estimated reliably based on this system.
3. The authors should validate their system by sorting the different populations of cells (mCherry+/GFP-, mCherry-/GFP-, mCherry+/GFP+) and performing NGS on each one of them to make sure that the profiles of indels and deletions actually follow their flow analysis. Furthermore, the authors should actually measure the sizes of their large deletions using some long read NGS techniques on the sorted populations of cells. These are very important issues not addressed in the manuscript.
4. The authors designed the targeting gRNA such that deletions >250bp would inactivate both proteins. Such deletions are called “large”. The ratio between “small” indels and “large” deletions serves as the basis for distinguishing the contributions of different repair pathways. However, the reason for choosing the 250 bp threshold to distinguish different repair pathways is not described in the paper.
5. I was very surprised to not find a single p-value estimate of significance of the many effects described in the manuscript. For example, there are no estimates of significance of the observed effects in the knockouts of specific genes (Figure 2), in the genetic screen (Figure 4) or in the treatments with specific drugs (Figure 3). The authors must provide these estimates, and they have to be based on analyses of these various knockouts and treatments in different biological replicas, ie different batches of cells.
6. Related to the above, the authors do not provide any information on the stability of their method in different biological replicas. They only mention 3 technical replicas in Methods. The information on biological replication is critical to understanding the robustness of their method and validity of the conclusions.
7. I am somewhat surprised that the authors did not develop cell lines stably expressing the gRNA. Transfection of gRNA complexes can be quite variable and also lead to non-physiologically high amounts of the synthetic RNAs in cells that in turn can lead to all sorts of artifacts (PMID 26697058). For example, there is a big difference between the two WT samples in Figure 2A. This is another reason to show how reproducible the system is in different biological replicas. Also, a stable cell line would be an improvement since it is more likely to stabilize performance of this method across multiple laboratories if it is to be adopted.
8. Related to the above, the authors chose immortalized non-cancerous cell line HEK293. Would it not be more appropriate to test this system in cancerous cell lines since they are more likely to be used as drug targets? I would imagine that the reason HEK293 cells are used here is because they

are easy to transfect, but this significantly limits applicability of this method. This is another reason to generate cell lines stably expressing the gRNA(s).

9. The attempt to use this method to build a predictor based on machine-learning approach random forest to classify different drugs is interesting. However, I see a number of issues with it:

9.1. I can not follow the "Random forest model – bioinformatics pipeline" section in the Methods. It definitely needs to be improved for clarity and logic. Among others, the authors need to explain why they remove data points similar to DSB controls and what are those DSB controls. What is "N" in the Table 1? Number of wells? And so on...

9.2. The performance of this model in the text needs to be described better. Did the authors validate it only on drugs they tested? What if they used drugs not used to build the model? If this method is to be applied broadly, this question has to be addressed.

Some additional comments:

1. In Material and Methods section, the authors wrote: "Single-cell suspensions were analyzed, and cells with strong eGFP (488-440 530/30) and mCherry (561-610/20) signals were sorted using FACSaria I cell sorter (BD 441 Biosciences) to enrich cells harboring the reporter." Is the reporter cell line represented by clonal or mixed population? Do all the cells have the same single genomic locus insertion? This point is not very clear.

2. Inhibitor studies in Figure 3. Do the authors have evidence that these drugs work as expected at these concentrations in this cell line?

3. In the Discussion, the authors state that "Our results are consistent with a recent study describing large deletions ranging from 250 bps to 6 kbs to occur more than 20% of the cases upon Cas9 induced DSBs in mouse embryonic stem cells and in RPE-1 cells (ref 32)." The reference for this sentence seems to be incorrect.

Reviewer #3 (Remarks to the Author):

In this manuscript the authors assembled a fluorescence-based reporter that allows simultaneous monitoring of end-protection and end-resection DSB repair processes, and showed that it enabled classification of a panel of DDR-targeting small molecules / DDR genes in choice of DSB repair pathways. The authors went on to demonstrate that inhibiting NER led to increased efficiency in SSTR. Overall the manuscript is clearly written and conclusions were supported by experimental findings.

I have a few comments that the authors may consider when revising the manuscript -

1) While it appears that the eGFP-targeting gRNAs are efficient in inducing DSBs at the integrated cassette (Figure 1b; From the scatter plot one would estimate that DSBs were induced in at least 92% of cells) the genomic cleavage assay suggests otherwise (Supp Figure 1b). The authors should explain this apparent discrepancy.

2) It remains unclear how "error-free" repair is effected upon DSB induction in CAT-R. The authors described HR as one possible means but is there a homologous template to allow HR to take place? What is the contribution of small InDels that took place in the multiplicity of 3 nucleotides that would presumably preserve eGFP?

3) The statement that "These results agree well with the idea that in the presence of PARP, the DSBs are repaired by alternative NHEJ pathway, which may contribute to the formation of large deletions due to extensive end-resection" is confusing given that "PARP inhibitors led to a reduction of small InDels and an increase of large deletions on average by 4% (+/- 1.6) at 50nM (Figure 3f).

4) CAT-R does not appear to be very sensitive in detecting change in "error-free" repair when established HR factors are inactivated. Perhaps the authors can discuss the limitation of the DSB repair reporter.

5) The authors showed that pre-treatment with the DDR gene-targeting small molecules affect cell cycle distribution. (How) Does change in cell cycle distribution affect DSB induction per se?

6) I see that there are multiple "WT" plots in Figure 2a, 2c and 2e. Are they "WT" representative plots?

Response to reviewer comments

NCOMMS-19-37771-T

A scalable CRISPR/Cas9-based fluorescent reporter assay to study DNA double-strand break repair choice

Paris Roidos^{1#}, Stephanie Sungalee^{2,3#}, Salvatore Benfatto¹, Özdemirhan Sercin¹, Adrian M. Stütz², Amir Abdollahi⁴, Jan Mauer¹, Frank T. Zenke⁵, Jan O. Korbel², Balca R. Mardin¹

We would like to thank all reviewers for their ideas and helpful suggestions. Additionally, we thank the reviewers for their positive and supportive comments. We were delighted to see that the reviewers were overall positive about our method, acknowledging the potential importance and novelty of the approach.

They raised some important points and had helpful suggestions for improving the manuscript. We now present a revised manuscript that addresses all the reviewer comments.

Based on the suggestions of the reviewers, we have:

- Performed long-read sequencing with Oxford Nanopore Technology to characterize the profile of large deletions.
- Engineered our first model cell lines to stably express the gRNA to induce a double-strand break, and we provided an additional workflow for integration of our reporter with the use of the AAVS1 safe harbor system into cancer cell lines.
- Recapitulated the finding that the NER pathway plays an active role in single-strand template repair in an additional cell line.
- Improved the data analysis and the representation of the statistical tests.
- Finally, we put substantial effort into addressing the interpretation of this method, especially to avoid any claims that our system can quantify "error-free" repair. We emphasized that our system is best used as a ratiometric reporter to measure error-prone repair.

We hope the reviewers share our excitement about our reporter system and the new concepts described in the revised manuscript. The changes in our revised manuscript are indicated with bold and italics. Please find below our point-by-point response to reviewers' comments.

Reviewer1:

- 1. The Authors need to clearly mention the superiority of the proposed assay system from other DSB repair assays already present. Additionally, multiple cell lines should be tested as clearly HEK293 and RPE1 showed differences in basal level.**

We thank the reviewer for this comment. Following the reviewer's suggestion, we added the following paragraph in our introduction (page 3, line 58):

- *“However, most of these reporters are limited to interrogate one repair pathway at a time, which may not be suitable to characterize the complex DSB repair response. Additionally, DSB events that can be induced and tracked by these reporters range from 1 to 25 % of the population. Thus, a robust and efficient reporter that allows capturing various responses to DSBs is still missing.”*

Moreover, following the 'reviewer's comment, we established a new system, based on directed integration into the AAVS1 genomic locus, allowing us to integrate the CAT-R construct in additional cell lines. We demonstrate the success of this approach by the integration of CAT-R in NCI-H358 cells. In this case, we also detected both small InDels and large deletions generated upon a Cas9 induced DSB, similar to the phenotype of the reporter in RPE-1 and HEK293 cells. These results are presented in Supplementary Figure 7a-b in our revised manuscript.

We note, however, based on the genetic background of a given cell line, that the ratios of small InDels/large deletions will vary between cell types. Indeed, we detected small changes in the ratios of RPE-1^{CAT-R} and HEK293^{CAT-R} cells, which may stem from the differences in their cell cycle profiles, as we comment on in the manuscript (page 5, line 110).

- *“The small difference in the ratio between these two model cell lines may be explained by the slight changes in the cell-cycle profile, whereby RPE-1 cells spend a longer time in the G1 phase.”*

In the case of the NCI-H358 cell line, we observed a higher rate of small InDels compared to HEK293 or RPE-1 cells. The genetic background of cancer cell lines can affect DSB repair choice; however, specific deficiencies of cancer cell lines guiding this choice can be challenging to decode from genome sequencing. We could speculate that CAT-R can eventually be used as a tool to understand the DSB repair choices in cancer cell lines. This exciting possibility, despite being beyond the scope of this manuscript, is now discussed in detail in our revised manuscript (page 14, line 437).

- *“As a proof of principle, we demonstrated the potential use of this approach by integrating this construct in the NCI-H358 lung cancer cell line (Supplementary Figure 7b). This approach can be used to study how DSB repair choices are made and how they can be affected in various cancer cell lines.”*

- 2. The double-positive (mCherry and eGFP) population shows both untransfected and error-free repair active cells. This might affect the interpretation of the overall repair capacity and repair choice of the cells. The author needs to state how they have overcome this issue clearly.**

We thank the reviewer for pointing this out and agree that our system is not well suited to detect error-free repair. In the revised version of our manuscript, we made several changes in the text as well in the figures to remove any ambiguity

regarding the interpretation of the overall repair capacity and repair choice of the cells with our system. In our revised manuscript, we state:

- "We anticipate that the population with intact mCherry and eGFP sequences is likely a combination of untransfected cells, and, DSBs that underwent error-free repair. However, the specific events that lead to the mCherry+/GFP+ population cannot be resolved with this reporter. For this reason, we focused on the two populations that represent the error-prone repair of the DSB (Figure 1b)" (page 5, line 97).
- "Due to the high efficiency of DSBs introduced by the CRISPR/Cas9 system, CAT-R can simultaneously track the formation of small InDels and large deletions in a ratiometric way with high resolution. We stably introduced CAT-R at a single genomic locus in two non-cancerous cell lines with intact DNA repair pathways. We demonstrated that we could quantify the rates of the populations that underwent error-prone DSB repair while with this reporter, an error-free repair cannot be directly quantified. This is because the error-free population represents a mixture of potential outcomes such as (i) untransfected cells, (ii) cells that underwent homologous recombination, or (iii) some small InDels that are multiplications of 3 nucleotides, or products of error-free NHEJ." (page 12, line 346).

In addition, we changed the annotation "error-free" to "untransfected/error-free repair", throughout the text and figures.

Furthermore, we emphasized the ratiometric aspect of the system that allows the distinction of small InDels to large deletions by stating:

- "In this study, we developed and utilized a ratiometric fluorescent reporter system..." (page 4, line 63).

3. Between the large deletions (double negative population) Vs. InDels (eGFP negative) there will be a range of other events, is the assay sensitive enough to measure the array of fluorescence change? This might be helpful in understanding the extent of DDR protein's influence on pathway choice.

To better understand the nature of the repair products and to also address a comment from reviewer2 (please see reviewer2, point 3), we performed long-read sequencing based on the Oxford Nanopore Technology (ONT) on the sorted error-prone populations. These data are now presented in Figure 1f.

The combination of short and long-read sequencing of amplicons around the eGFP cut site allowed us to understand better the repair products and the range of events that occur upon a Cas9 induced DSB. Based on short-read (Illumina) sequencing, we have demonstrated that 98% of small InDels comprise mainly of events smaller than 30 bp. Consistently our ONT-based long-read sequencing analyses support these findings. Additionally, we noticed that approximately 4% of the events that are still classified as "small InDels" represent 150 bp deletions, which are likely products of alt-EJ based mechanisms. Large deletions, however, are products of more extended end-resection events, as demonstrated by the ONT data. These data are described and discussed in our revised manuscript (page 6, line 131). We believe that with CAT-R, we capture the vast majority of the events that can occur due to Cas9 induced DSBs.

- 4. Using a CRISPR library to assess the role of different repair proteins on DSB repair choices is a robust way to test the system and identify new dependencies on pathway choice. However, the authors need to address the transfection efficiency of both gRNA to eGFP and CRISPR library to cells. From that screen, authors found that the NER pathway plays an active role in CRISPR-knock in ability (although clear data is missing), this data has a potential and should be expanded and studied with more positive and negative controls to make a claim.**

We thank the reviewer for their suggestions. In our revised manuscript, we provided additional data demonstrating the efficient transfection of the gRNAs and our library into cells. Furthermore, we improved the section on the NER pathway playing an active role in SSTR by additional experiments in two independent cell lines. In particular:

1. *In Supplementary Figure 1. CAT-R optimization (a) and (b), we compare the use of different gRNA formats to transfection efficiency. We demonstrate the superiority of the RNA format over the plasmid DNA efficiency, regarding with regards to timing and cleavage.*

2. *In Supplementary Figure 6. CAT-R as a reporter of single-strand template repair. (a) we demonstrate the CRISPR library transfection efficiency per plate. We note that our arrayed library is delivered in 10x 96-well plates. In each plate, we included two positive controls for transfection and six negative controls (non-targeting scrambled gRNA). As a positive control, the essential POLR2A gene is used with its phenotype to be a reduction in cell viability. Therefore, if the number of viable cells in those wells is low (< 1.500 counts that correspond to >85 % transfection efficiency), then we consider this plate as successfully transfected. To clarify this point, we added the following text in the Material and Methods section:*
 - *"Previously, we showed that targeting POLR2A can serve as a positive control of transfection efficiency³⁵. POLR2A is an essential gene for the survival of a cell. Therefore, plates in which POLR2A gRNA transfected wells contain more than 1.500 cells (indicating lower than 85% transfection efficiency) were removed from the subsequent analysis due to poor transfection conditions. The data from the two replicates were normalized to gRNA:eGFP WT control and then averaged per gene." (page 21, line 704)*

3. *In our revised manuscript, we now devote a new Figure (Figure 5) to emphasize the potential role of NER in increasing the efficiency of CRISPR-knock ins by providing data from additional experiments as well as a model of how these events can be resolved:*
 - a. *We recapitulated the results of NER deficiency in another cell line (RPE-1).*
 - b. *We included DNA-PK inhibitor and genetic depletion as positive controls.*
 - c. *We suggest a mechanism of Cas9-mediated DSB repair choice that explains the possible positive contribution of NER deficiency to CRISPR knock-ins. Please refer to Figure 5. Knocking-out NER increases the chances of a successful knock-in. (c)*
 - d. *Moreover, we included the following text in the discussion:*

"In addition, it has been demonstrated that the gRNA sequence is bound to the antisense strand as an RNA:DNA hybrid and a 5' to 3' flap is generated at the non-targeted/sense sequence^{55,56}, which can be processed by NER. Knocking out NER may thus increase the chances of a successful knock-in via SSTR/HDR." (page 13, line 413)

In summary, we believe that we provided additional evidence and discussion to emphasize the role of NER in regulating SSTR-based knock-ins. We thank this reviewer for their comments and ideas for our manuscript.

Reviewer2:

- 1. The approach has one fundamental flaw – the DNA molecules digested by Cas9 and repaired error-free can also be cleaved again until they are repaired with errors such that they can longer be substrates for the Cas9 enzyme. Therefore, this is not a very good system to estimate the fraction of error-free repair.**
- 2. Related to the point above, it is impossible to distinguish error-free repair and uncleaved DNA, further arguing that error-free repair can not be estimated reliably based on this system.**

We agree with the reviewer, and although we did not intend to make any claims on the ability of the CAT-R reporter to reliably quantify error-free repair, in our revised manuscript, we modified the text and the figures to avoid any misinterpretations of the efficiency of the error-free repair. We now instead emphasize the ratio between small InDels to large deletions since this is the primary outcome of the ratiometric reporter. We also discuss in more detail the limitations of this system. Nonetheless, we believe that this reporter has several advantages:

- 1. CAT-R has very high efficiency of DSB induction; thus, it can be used even to detect minor changes in DSB repair choice.*
- 2. CAT-R has the potential to be implemented in chemical and genetic screens, as demonstrated in our manuscript.*
- 3. To our knowledge, CAT-R is the first reporter that can simultaneously track end-protection and end-resection mediated repair.*

In our revised manuscript, we made these points clearer while acknowledging the potential drawbacks of the current system. More details on specific amendments to the revised manuscript can be found in our answer to reviewer1, point 2.

- 3. The authors should validate their system by sorting the different populations of cells (mCherry+/GFP-, mCherry-/GFP-, mCherry+/GFP+) and performing NGS on each one of them to make sure that the profiles of InDels and deletions actually follow their flow analysis. Furthermore, the authors should actually measure the sizes of their large deletions using some long-read NGS techniques on the sorted populations of cells. These are very important issues not addressed in the manuscript.**

We thank the reviewer for this suggestion. In the revised manuscript, we performed long-read PCR sequencing (8.5 kb fragments) with Oxford Nanopore

Technology (ONT) not only on the sorted "large deletion" population but also "small InDels" population in two independent biological replicates as well as an uncut sample and an unsorted DSB sample. The detailed analyses of these experiments are presented in Figure 1f and Supplementary Figure 1k, l, and discussed on page 6 of the revised manuscript. In summary, based on these data, we can conclude that:

- 1. Large deletions that can be detected within the range of our PCR amplicon are better resolved in the sorted populations.*
- 2. Although we may miss larger than 8.5kb deletions due to the size of our PCR product, we detect a wide range of events that can be classified as large deletions ranging from 500 bp-8.1 kb with a median size of 4 kb.*
- 3. The majority of the resection events are upstream of the Cas9 break site consistent with the earlier reports suggesting that upon DSB, Cas9 remains bound to the DNA.*
- 4. Although the base-pair resolution of the ONT data is not ideal due to the nature of the current technology, we detected microhomologies around large deletions, which shed light on how these large deletions can be repaired.*

Taken together, we believe that these data indeed strengthen the manuscript by providing valuable information about the characterization of repair events at the break site that can be identified by CAT-R.

All these points are presented and discussed in the revised manuscript (page 6, line 128):

- "Although in most cases, the repair of the Cas9-induced DSBs is expected to result in small InDels due to the action of c-NHEJ or alt-EJ, recent studies also suggest that large deletions can occur frequently. Since these larger events cannot be observed by short-read sequencing, we performed long-read sequencing based on the Oxford Nanopore Technologies (ONT) to detect the composition of large deletions events up to 8.5 kb, generated at the target site in unsorted as well as sorted populations. The median size of the deletions we observed was 4 kb, with a maximum size of 8.1kb. Interestingly, most of the deletions we observed were larger than 3 kb, with the most common class of deletion events to be between 5000 - 8100 bp having a frequency of 22% (Figure 1f). These results suggest that large deletions as a product of end-resection events are frequent upon DSB break induction. In addition, we observed, based on the ONT data, that the resection events are asymmetric to the target site, with the majority of the resection events to take place upstream of the PAM site (Supplementary Figure 1k). These large deletions at least partly may be repaired via microhomology-mediated repair as we observed frequent microhomologies at the break sites, consistent with the current literature suggesting microhomology-mediated repair upon Cas9 induced large deletions (Supplementary Figure 1l). Overall, we demonstrate that based on the color of the populations upon a DSB, CAT-R allows the determination of the frequency of large deletions in addition to small InDels in a quick and robust manner."*

- 4. The authors designed the targeting gRNA such that deletions >250bp would inactivate both proteins. Such deletions are called “large”. The ratio between “small” InDels and “large” deletions serves as the basis for distinguishing the contributions of different repair pathways. However, the reason for choosing the 250 bp threshold to distinguish different repair pathways is not described in the paper.**

We updated this section in our revised manuscript based on the new evidence we gathered from both Illumina and ONT data. We now state that based on the design of the construct, the distance between the cutting site and the mCherry sequence is 420 bp. For this reason, we hypothesized that a minimum of 420 bp should be affected to observe an effect on both fluorescent proteins.

Indeed, based on both Illumina and ONT data, the majority of our small InDels are less than 30 bp, with a max of 150 bp deletions. We, however, agree that this point may not be apparent in the text. For this reason, in the revised manuscript, we added the following sentence "(the mCherry sequence is 420 bp away from the cutting site)" and we adapted Figure 1. The Color Assay Tracing Repair (CAT-R) reporter system, accordingly.

- 5. I was very surprised to not find a single p-value estimate of the significance of the many effects described in the manuscript. For example, there are no estimates of the significance of the observed effects in the knockouts of specific genes (Figure 2), in the genetic screen (Figure 4) or in the treatments with specific drugs (Figure 3). The authors must provide these estimates, and they have to be based on analyses of these various knockouts and treatments in different biological replicas, ie different batches of cells.**

We thank the reviewer for raising this critical point and apologize for not including statistical testing in the first place. In our revised manuscript, regarding the p-value estimates, we performed a multiple comparisons test for each data set (knockouts of specific genes and genetic screen) to examine whether the derived populations are significantly different from the wild-type (WT). We have included the following text in the Material and Methods section and the appropriate annotations in the respective figures:

- "The estimates of significance were determined using a mixed-effect model analysis for multiple comparisons. Every sample is compared to the WT control with the mean of each CAT-R population to be compared with the respective control mean. Each P value is adjusted to account for multiple comparisons using statistical hypothesis testing with the Dunnett test. Random effects with zero SD were excluded from the model. For data analysis and visualization, the GraphPad Prism 8 was used." (page 20, line 658)*

Additionally, regarding the treatments with specific drugs, we included a supplementary table (Supplementary Table 2.) documenting the dose-response statistics for the drug screens.

6. Related to the above, the authors do not provide any information on the stability of their method in different biological replicas. They only mention 3 technical replicas in Methods. The information on biological replication is critical to understanding the robustness of their method and the validity of the conclusions.

We apologize for not including clear statements about the number of biological replicates used throughout this manuscript. The strength of our method is its compatibility with high throughput flow cytometry, and we performed several independent biological and technical replicates in each experiment.

In our revised manuscript, in each figure, two types of "n" are used to describe sample sizes.

n = all replicates (including technical), and this is used only for the CAT-R phenotype as a system and for the genetic manipulations.

N = all biological replicates (independent experiments), and this is used for cell viability, colony formation assay, siRNA experiments, cell cycle analysis, qPCR, drug screen.

We clarify all these points in our revised manuscript and have included the following text in the Material and Methods section:

- *"We note that we define biological replicates as completely independent experiments carried out on different days with a different batch of materials. In each biological replicate, we always include three technical replicates that represent, e.g., 3 wells of a 96-well plate (N represents the number of biological replicates). Results from the reporter are presented as min to max, showing all points of all experiments (n represents the number of all technical replicates)." (page 19, line 650)*

To provide evidence for the robustness of our method, in our revised manuscript we combined data from all the independent experiments that we carried out throughout the manuscript (in the absence of any genetic or chemical manipulation) and plotted the ratio between small InDels and large deletions (x-axis) vs. the untransfected/error-free population (y-axis). The ratios of small InDels to large deletions are very similar regardless of the transfection efficiencies. (average 1.18, sd \pm 0.23 for HEK293 and 1.24 \pm 0.20 for RPE-1). We think that, especially when the transfection efficiencies are above 50 % (which can be easily achieved in many cell lines using gRNAs), the CAT-R system can be reliably used to quantify the DSB repair choices. This new figure highlighting the robustness of the method is presented in Supplementary Figure 1g.

Additionally, we have included the following text in the Material and Methods section to explain better the efforts to optimize and ensure the system performance and reproducibility:

- *" Specifically, for examining the robustness of the method, 151 biological experiments were performed with HEK293^{CAT-R} and 56 biological experiments with RPE-1^{CAT-R}. Box and whiskers plots are used from min*

to max with median value to be annotated and values to be normalized to gRNA:eGFP WT control.” (page 20, line 655)

- 7. I am somewhat surprised that the authors did not develop cell lines stably expressing the gRNA. Transfection of gRNA complexes can be quite variable and also lead to non-physiologically high amounts of the synthetic RNAs in cells that in turn can lead to all sorts of artifacts (PMID 26697058). For example, there is a big difference between the two WT samples in Figure 2A. This is another reason to show how reproducible the system is in different biological replicas. Also, a stable cell line would be an improvement since it is more likely to stabilize the performance of this method across multiple laboratories if it is to be adopted.**

We thank the reviewer for this valuable comment, and indeed, our intention is to generate a system that can be used across multiple laboratories. For this reason, we have tested several adaptations of CAT-R and provided data derived from two cell lines with integrated gRNA and Cas9 in Supplementary Figure 7. While we agree with the reviewer that an integrated gRNA might be advantageous for eliminating the need for transfecting gRNAs, we would like to express our concerns about the leakiness that every inducible system has. Although we use the optimized inducible promoters and have tested several tetracycline-free sera, when we integrated the gRNA into our cells, we observed unwanted DSB events even in the absence of induction. This likely is due to the highly efficient CRISPR/Cas9 system, where even a few molecules of Cas9 complexed with the existing gRNA can be sufficient to induce a double-strand break. Accordingly, in the Supplementary Figure 7 c-f in our revised manuscript, we demonstrate in both HEK293 and RPE-1 cell lines, although Cas9 expression without Doxycycline induction is undetectable by immunoblotting, unintended DSBs are induced at high levels in the absence of Doxycycline. Nonetheless, all these cell lines will also be made available upon request.

In addition, we provided data demonstrating that the synthetic gRNA transfection is highly efficient in different cell lines, consistent with our recent publication (PMID: 31885201). Moreover, the synthetic gRNAs are complexed with the tracrRNA, and this complex is highly stable since they contain chemical modifications that protect them from degradation by cellular RNAses. This approach is thus unlikely to cause any unwanted side effects in cells (PMID: 27374403). Finally, we would like to point out that gRNA transfection has a central advantage compared to stable integration since it provides flexibility in choosing different gRNAs that can be combined with the CAT-R system without the need to generate new cell lines for each gRNA.

While we agree with the reviewer about the advantages of gRNA integration together with Cas9, we would again like to emphasize that, in many cases, this will give rise to unintended DSBs. By keeping Cas9 and gRNA separated, we wanted to make sure that the reporter system is not initiated until the Cas9 protein and the gRNA expressed together. In the revised manuscript, these points are discussed in the discussion (page 14, line 440):

- *"Additionally, gRNAs, together with Cas9, can also be stably expressed under an inducible promoter to induce a DSB. While this system can be*

advantageous to allow the study of DNA repair in model organisms, we also note that in such systems, unintended DSBs can be induced due to leakiness of the inducible promoters (Supplementary Figure 7d-f) and should be used with caution".

- 8. Related to the above, the authors chose immortalized non-cancerous cell line HEK293. Would it not be more appropriate to test this system in cancerous cell lines since they are more likely to be used as drug targets? I would imagine that the reason HEK293 cells are used here is because they are easy to transfect, but this significantly limits applicability of this method. This is another reason to generate cell lines stably expressing the gRNA(s).**

In this manuscript, we describe a reporter system that can quantify the choices in DSB repair. To study this, we needed to establish our system in immortalized non-cancerous cell lines such as HEK293 and RPE-1 since they are unlikely to impact the CAT-R readouts due to their genetic background. After establishing the characteristics of the system, demonstrating the robustness and the use of this system in both chemical and genetic screens, we believe that we established a baseline of DSB repair choices in genetically stable cell lines that are altogether presented in this manuscript. We agree with the reviewer that an essential next step would be to understand how DSB repair choices are made and how they can be affected in various cancer cell lines and although that goes beyond the scope of this current study, we now describe and provide an alternative approach to generate cell lines integrating the CAT-R system using the adeno-associated virus site 1 (AAVS1, please see also reviewer 1 point 1). In this system, the CAT-R construct can be integrated into a safe harbor genomic locus. We demonstrate the potential use of this approach by integrating this construct in the NCI-H358 lung cancer cell line. In our revised manuscript, we included a section in which we describe the potential adaptations of the current CAT-R system (e.g., AAVS1 system or stable integration of gRNAs). We hope that these additional data will not only help the scientific community to make an informed decision about which system they can best use in combination with CAT-R but also show how robust our system is in detecting DSB repair choices.

- *"CAT-R reporter can be utilized in several ways and can be adapted in additional model systems to understand how DSB repair choices are made in various cancer cell lines. One alternative approach is via integration of the CAT-R reporter in Cas9-expressing cell lines using the adeno-associated virus site 1 (AAVS1) (Supplementary Figure 7a). In this efficient system, the CAT-R construct can be inserted into the same chromosomal location as a single stable copy. As proof of principle, we demonstrated the potential use of this approach by integrating this construct in the NCI-H358 lung cancer cell line (Supplementary Figure 7b). This approach can be used to study how DSB repair choices are made and how they can be influenced in various cancer cell lines." (page 13, line 431).*

Moreover, we have included a new figure; please refer to Supplementary Figure 7. Expanding the utility of CAT-R (a), where we describe the workflow of generating CAT-R stable expressing cell lines.

9. The attempt to use this method to build a predictor based on machine-learning approach random forest to classify different drugs is interesting. However, I see a number of issues with it:

9.1. I can not follow the “Random forest model – bioinformatics pipeline” section in the Methods. It definitely needs to be improved for clarity and logic. Among others, the authors need to explain why they remove data points similar to DSB controls and what are those DSB controls. What is “N” in the Table 1? Number of wells? And so on...

We provided a more detailed explanation of the machine learning approach in our revised manuscript to clarify the points raised by the reviewer. For instance, in the Material and Method section, we included how the data is analyzed in greater detail (page 20, line 671):

- *"The raw FACS data were transformed into a vector before being used as input for the Random Forest (RF) model development. Initially, for each sample, the 2D kernel density was computed from the raw FACS data, and the resulted plot was converted into a 100 x 100 pixels image. The image was then flattened into a 10.000 elements vector.*

To avoid false predictions, we only included samples where the effect of the compounds was significantly different from the DSB controls (in the absence of any drugs). To identify such cases, we computed the statistical distances of each sample from the controls per 96-well plate using the Kolmogorov–Smirnov test, and we calculated the average. As a reference, we used the distribution of the average statistical distances within the controls; its 95% confidence interval (CI) upper endpoint was used as a cut off to remove samples with a similar phenotype to the control.

Then the dataset was split randomly into training and test sets using an 80/20 ratio, preserving the overall class distribution of the data. The training set was used to train an RF model. The hyperparameter tuning was performed randomly, partitioning the data in 5 equal-sized subsamples of which one was retained as a validation set and the others as a train set. The process was repeated five times so that each subsample was used as a validation set (5-fold cross-validation), then the model showing the highest accuracy was selected. Inside this process, the minority drug class was randomly sampled to be the same size as the majority drug class. To assess the final model, we predicted the classes of the test set and generated the confusion matrix to calculate the performance of the model. The modeling was performed in R using the caret and the ranger packages."

In addition, we clarified the explanations in Table1.

9.2. The performance of this model in the text needs to be described better. Did the authors validate it only on drugs they tested? What if they used drugs not used to build the model? If this method is to be applied broadly, this question has to be addressed.

We thank the reviewer for raising this important point. In this study, we developed a Random Forest model to demonstrate how the CAT-R data can be

used to assign novel pharmacological compounds to known drug classes. We chose to develop a Random Forest model as a simple and widely used algorithm that increases the predictive power and helps to prevent overfitting of the data. We applied this method to classify (predict) novel compounds to a dataset of four major drug classes of enzymes that are responsible for DSB repair. Due to the nature of this predictive model, if a drug-class that the model is not trained for is used, then the model will be forced to classify the unknown drug-class to one of the four drug-classes that it is trained for. For this reason, while our model has a very high specificity and sensitivity to predict "novel" compounds of the four major classes of DNA repair inhibitors (ATM, ATR, DNA-PK, and PARP), it will not be able to predict any other inhibitor class due to the lack of data to train the model. Therefore, this model is designed to predict the efficiency of novel compounds of unknown efficiency that may be designed based on similarities to existing compounds, for which experimental validation can be missing to interpret their efficiencies. In this case, instead of laborious kinase assays and several other experimental strategies, CAT-R can be used as a simple method to predict their activity. In addition to predicting novel compounds that belong to these four major classes, we anticipate that in the future, CAT-R can be used to train on other inhibitor classes, and this model can be extended. In our revised manuscript, we present and discuss these results in:

- "To utilize CAT-R for potential drug-screening purposes, we built a machine-learning-based model that can predict the class of an unknown compound based only on its CAT-R phenotype. Therefore, we trained a random forest (RF) model with a dataset of CAT-R phenotypes (2.443 samples) from known compounds that belong to these four major classes of compounds and built a reference model." (page 9, lines 270)

And

- "This platform could provide further information on DDR kinase or PARP inhibitor drug discovery, serving as a tool to identify more selective inhibitors. We also developed a machine learning-based strategy to help classify unknown compounds in HEK293^{CAT-R} cells, though we note that the model should be reapplied to additional cell lines or different inhibitor classes to adapt to the responses in each cell line and inhibitor class, respectively". (page 13, line 394)

Moreover, we added a plot that accommodates the results of the RF-based model; please refer to Figure 3. (f) A platform to screen important DNA damage repair inhibitors.

Some additional comments:

1. In Material and Methods section, the authors wrote: "Single-cell suspensions were analyzed, and cells with strong eGFP (488-440 530/30) and mCherry (561-610/20) signals were sorted using FACS Aria I cell sorter (BD 441 Biosciences) to enrich cells harboring the reporter." Is the reporter cell line represented by clonal or mixed population? Do all the cells have the same single genomic locus insertion? This point is not very clear.

To address the reviewer's concerns, we included more information in the Material and Methods section as follows:

- "To generate cells that express the reporter as a single stable copy, the FLP recombinase methodology (Flp-InTM, InvitrogenTM) was used for HEK293 and RPE-1 cell lines. More specifically, the Flp-InTM T-RexTM system (InvitrogenTM) that allows tetracycline-inducible expression was

used only for the RPE-1 cell line. For the NCI-H358 cell line, the AAVS1 safe harbor targeting system was used according to the manufacturer's protocol (System Biosciences).

All cell lines contain the reporter at a single genomic locus. In the case of FLP integration, the transfected cells were selected in the presence of 500 µg/µl of neomycin for four days, and a mixed population was generated. In the case of the AAVS1 methodology, the transfected cells were sorted 1-2 weeks post-transfection. After expansion, single-cell suspensions from all cell lines were analyzed, and cells with strong eGFP (488-530/30) and mCherry (561-610/20) signals were sorted using FACS Aria I cell sorter (BD Biosciences) to enrich cells harboring the reporter." (page 15, lines 465)

2. Inhibitor studies in Figure 3. Do the authors have evidence that these drugs work as expected at these concentrations in this cell line?

In this study, we used validated drug compounds. In our revised manuscript, to make this point clear, we provide several citations in the section "CAT-R-based screening of clinically relevant DDR inhibitors" as also exemplified below.

Nonetheless, for a subset of DNA-PK, ATM, and ATR inhibitors, we performed quantitative immunoblots against the target protein (phosphorylated/total) as well as their downstream targets. (Supplementary Figures 3, 4), please also see the Material and Methods section "Drug target validation" (page 17, line 567).

Moreover, for the PARP compounds, we provide evidence that these compounds are functional by performing colony formation assay on isogenic BRCA proficient and deficient cells. These data are provided in Supplementary Figure 5 and described in the Material and Methods section "Colony formation assay" (page 18, line 579).

For additional information, please refer to the citations that we have included in the revised manuscript below:

Fokas, E. et al. Targeting ATR in vivo using the novel inhibitor VE-822 results in selective sensitization of pancreatic tumors to radiation. Cell Death Dis. 3, e441-10 (2012).

Sun, Q. et al. Therapeutic implications of p53 status on cancer cell fate following exposure to ionizing radiation and the DNA-PK inhibitor M3814. Mol. Cancer Res. 17, 2457–2468 (2019).

Zenke, F. T. et al. Pharmacological inhibitor of DNA-PK, M3814, potentiates radiotherapy and regresses human tumors in mouse models. Mol. Cancer Ther. (2020). doi:10.1158/1535-7163.MCT-19-0734

Degorce, S. L. et al. Discovery of Novel 3-Quinoline Carboxamides as Potent, Selective, and Orally Bioavailable Inhibitors of Ataxia Telangiectasia Mutated (ATM)

Kinase. J. Med. Chem. 59, 6281–92 (2016).

3. In the Discussion, the authors state that "Our results are consistent with a recent study describing large deletions ranging from 250 bps to 6 kbs to occur more than 20% of the cases upon Cas9 induced DSBs in mouse embryonic stem cells and in RPE-1 cells (ref 32)." The reference for this sentence seems to be incorrect.

We apologize for this mistake. The correct reference has been included and is:

Kosicki, M., Tomberg, K. & Bradley, A. Repair of double-strand breaks induced by CRISPR–Cas9 leads to large deletions and complex rearrangements. Nat. Biotechnol. 36, (2018)."

We thank the reviewer for their engaging and valuable suggestions. We believe these new experiments have substantially strengthened the conclusions in our study.

Reviewer3:

- 1. While it appears that the eGFP-targeting gRNAs are efficient in inducing DSBs at the integrated cassette (Figure 1b; From the scatter plot one would estimate that DSBs were induced in at least 92% of cells) the genomic cleavage assay suggests otherwise (Supp Figure 1b). The authors should explain this apparent discrepancy.**

We thank the reviewer for this comment and for allowing us to elucidate the potential reason for this discrepancy: The nature of the Surveyor assay is known to underappreciate the cleavage efficiency. First, it is a PCR based assay that restricts the detection levels to the amplicon size; therefore, only small InDels can be detected. Secondly, this reporter assay depends on the enzymatic cleavage of heteroduplex DNA that is generated during the hybridization step. Thus, it can only generate cleavage when the editing events are different. For these reasons, in our case, the genomic cleavage assay is used only to show that the correct genomic locus is cleaved, we do not think the quantification of the genomic cleavage assay is as sensitive as CAT-R.

In our revised manuscript, nonetheless, we provided an additional genomic cleavage assay figure (Supplementary Figure 1b), demonstrating the cleavage efficiency in both cell lines comparing the different gRNAs formats. In addition, we made clear in the manuscript that this assay is used only to ensure that the gRNA targets the correct genomic locus.

“We confirmed the site-specific genome editing events based on an enzymatic mismatch cleavage assay in a time-dependent manner and confirmed the different repair products by microscopy.” (page 5, line 92)

- 2. It remains unclear how "error-free" repair is effected upon DSB induction in CAT-R. The authors described HR as one possible means but is there a homologous template to allow HR to take place? What is the contribution of small InDels that took place in the multiplicity of 3 nucleotides that would presumably preserve eGFP?**

We thank the reviewer for this question. The CAT-R sequence is integrated as a single stable copy of the genome of the cells. During replication, two newly synthesized chromatids are created that can serve as a homologous template to allow HR to commence. However, as described in the previous sections, we would like to refrain from making any comments about the HR activity with the CAT-R reporter, since the primary function of CAT-R is to report the ratio between small InDels and large deletions. Nevertheless, we adapted CAT-R with the use of an exogenous template in the form of ssODN to accurately quantify the activity of single-strand template repair (SSTR) events that are considered a subset of HR.

Having small InDels in multiple of 3 nucleotides is a valid argument for small InDels that may not affect the eGFP folding. Although it is currently unclear how many amino acid deletions would lead to misfolding of eGFP protein, based on

our NGS analysis for the small InDels, we can see that the maximum frequency of 3 nt deletions is 5 %, and this number is further decreased as the length of the deletions increase.

In the revised manuscript considering this, we added the following sentence describing the CAT-R reporter:

- *"We demonstrated that we could quantify the rates of the populations that underwent error-prone DSB repair while with this reporter, an error-free repair cannot be directly quantified. This is because the error-free population represents a mixture of potential outcomes such as (i) untransfected cells, (ii) cells that underwent homologous recombination, or (iii) some small InDels that are multiplications of 3 nucleotides, or products of error-free NHEJ." (page 12, line 350).*

- 3. The statement that "These results agree well with the idea that in the presence of PARP, the DSBs are repaired by alternative NHEJ pathway, which may contribute to the formation of large deletions due to extensive end-resection" is confusing given that "PARP inhibitors led to a reduction of small InDels and an increase of large deletions on average by 4% (+/- 1.6) at 50nM (Figure 3f).**

We apologize for the wrong expression. Since both the PARP1 KO and the PARP inhibition phenotype shows a reduction of small InDel formation and an increase of large deletions, we modified this sentence as follows:

- *"These results agree well with the idea that in the presence of PARP, the DSBs are repaired by the alternative EJ pathway, which may contribute to the formation of small InDels due to end-protection." (page 9, line 253).*

- 4. CAT-R does not appear to be very sensitive in detecting change in "error-free" repair when established HR factors are inactivated. Perhaps the authors can discuss the limitation of the DSB repair reporter.**

We have now changed our narrative to refrain from making any claims concerning homologous recombination events using CAT-R. Based on all the reviewers' suggestions, we have made careful comments on how CAT-R should be used and what its limitations are. The detailed responses and the amendments that are related to this section can be found in our responses to reviewer1, point2, and reviewer2, point 1.

- 5. The authors showed that pre-treatment with the DDR gene-targeting small molecules affect cell cycle distribution. (How) Does change in cell cycle distribution affect DSB induction per se?**

We thank the reviewer for this question. Indeed, specific cell cycle stages differentially affect DNA repair choice since most DNA repair pathways are active in specific points during the cell cycle progression. It is also well documented that inhibition or KO of essential DDR genes may have some direct or indirect effects on the cell cycle, as we also show in Supplementary Figures 3 & 5.

In this study, we are examining a Cas9-mediated DSB repair. However, the cut and repair process of Cas9 cleavage is not yet fully understood. To the best of

our knowledge, it commences 10-14 h post-transfection. In the meantime, the cell cycle progresses normally. We do not anticipate any effect on the induction of DSBs since there is no evidence showing that the activity of Cas9 is regulated by the cell cycle. In addition, we reach similar cutting efficiencies even when we use DDR inhibitors. For instance, the use of the ATM inhibitor AZD0156 leads to a slight increase in the G2/M phase (Supplementary Figure 3i); however, the transfection efficiencies between the AZD0156 treated samples, and the DMSO treated samples do not differ. The only difference that we observe with CAT-R is the change in the CAT-R phenotype, which is the ratio between the two error-prone populations. Thus, we do not expect any substantial impact on the DSB induction by the changes in cell cycle distribution.

6. I see that there are multiple "WT" plots in Figure 2a, 2c and 2e. Are they "WT" representative plots?

Experiments describing the observed effects in (i) the knockouts of specific genes, (ii) the genetic screen, and (iii) the treatments with specific drugs were accompanied by a "WT" control to allow for normalization to DSB efficiency. Since, in each experiment, we transfect our gRNA targeting eGFP, in different biological experiments, we reach slightly different levels of transfection. Although the level of transfection does not affect the response that we observe using our reporter (as explained in our response to reviewer2, point 6), we believe this is the most comprehensive way to demonstrate the controls that are used in these experiments.

In our revised manuscript, we made this point clear in the figure's legend:

- *"Representative flow cytometry analysis plots of HEK293^{CAT-R} cells 72 h post-transfection with the synthetic gRNA targeting the eGFP coding sequence in (a) PRKDC and XRCC4 KO cells, (c) pool of CRISPR/gRNA transfected cells, and (e) ATM and PARP1 KO cell lines, each compared to their representative WT controls."*

We thank the reviewer for their helpful comments and suggestions, and we appreciate the enthusiasm they showed for our manuscript. We believe that our amendments have substantially improved this manuscript.

REVIEWER COMMENTS

Reviewer #1 (Remarks to the Author):

The authors have addressed most of my comments.

Reviewer #2 (Remarks to the Author):

The manuscript has improved significantly and the authors have addressed a significant fraction of the queries. However, I still have the following issues.

1. The authors have presumably sorted the mCherry+/GFP- and mCherry-/GFP- populations and directly measured the distributions of the lengths of breaks in each sub-population using Illumina and Oxford Nanopore platforms. However, the authors do not explicitly state what populations were used for the sequencing neither in the main text nor the Methods part. They only use word "sorted" in the Figure 1 and Supplementary Figure 1 and therefore it is not exactly clear what was done. This analysis is at the core of validation of their high-throughput assay. As such, it should be emphasized and described in much more detail in the main text and methods. It would be also good to see a panel in a main figure that shows exact gating parameters used in this sorting-sequencing experiment.

2. In terms of the inducible Cas9 cell lines, the authors state that they cannot detect Cas9 expression without Doxycycline induction, while they still observe the unwanted DSBs events in the absence of induction and claim that those are due to the leakiness of the inducible promoters. I think the authors should provide more evidence to support this conclusion, and/or they should provide a more detailed clarification of the observed results. Furthermore, the Cas9-mediated site-specific DNA cleavage is at the core of their method and the authors should show that the increase in the breaks and increase in the fraction of the mCherry+/GFP- and mCherry-/GFP- populations is Dox-dependent.

3. The authors now include estimates of the statistical significance, however, they only use them in the context of yes or no answers as to the significance of their observations. The readers might want to look at the actual values to make their own determination about relative significance of various treatments. Therefore, I think the author should make a Supplementary Dataset with the actual p-values and %-ages for every sorted fraction for every drug treatment and gene knockout (eg, ATM, PARP etc) tested in this work.

4. The authors provide 3 supplementary tables, but they only mention one of them in the text.

Reviewer #3 (Remarks to the Author):

The authors have adequately addressed my critiques.

Response to reviewer comments

NCOMMS-19-37771-A

A scalable CRISPR/Cas9-based fluorescent reporter assay to study DNA double-strand break repair choice

Paris Roidos^{1#}, Stephanie Sungalee^{2,3#}, Salvatore Benfatto¹, Özdemirhan Sercin¹, Adrian M. Stütz², Amir Abdollahi⁴, Jan Mauer¹, Frank T. Zenke⁵, Jan O. Korbel², Balca R. Mardin^{1*}

We were pleased to see that all three reviewers acknowledge that our work was greatly improved, we thank them for their positive and supportive comments. Reviewer #2 raised few final points and we address these in our point-by-point response below. The changes in our revised manuscript are indicated with bold and italics. Based on the suggestions of the reviewer, specifically we have:

- Improved the illustration of our gating strategy for the sorting experiments.
- Provided a more detailed clarification of our results with regards to the potential pitfalls of the dox inducible systems when both gRNA and Cas9 are integrated in the cells.
- Included additional Source Data files for the (1) KO screen raw dataset including *p*-value calculations, the (2) Drug screen raw dataset.

Reviewer2:

1. The authors have presumably sorted the mCherry+/GFP- and mCherry-/GFP- populations and directly measured the distributions of the lengths of breaks in each sub-population using Illumina and Oxford Nanopore platforms. However, the authors do not explicitly state what populations were used for the sequencing neither in the main text nor the Methods part. They only use word “sorted” in the Figure 1 and Supplementary Figure 1 and therefore it is not exactly clear what was done. This analysis is at the core of validation of their high-throughput assay. As such, it should be emphasized and described in much more detail in the main text and methods. It would be also good to see a panel in a main figure that shows exact gating parameters used in this sorting-sequencing experiment.

We have sorted the mCherry+/GFP- and mCherry-/GFP- populations following the scheme presented in Figure 1b and proceeded with our NGS analysis. Following the reviewer's suggestion, we adapted our text in Results as well as in Methods sections as indicated below, we made small amendments to the Figure1 to indicate the two sorted populations more clearly and finally, we included an additional figure illustrating the gating strategy used in the long-read sequencing experiments in Supplementary Figure 1k.

*“...as well as sorted population, as indicated in Supplementary Figure1k”,
(page 6, line 153)*

“Three populations were defined and classified as: mCherry+/eGFP+, mCherry+/eGFP- and mCherry-/eGFP-. Populations were defined as negative when the fluorescent intensities were at least an order of magnitude lower than the median of each sample.” (page 15, line 510)

“For long-read sequencing, we analyzed a mixed cell population sample upon a DSB, and two sorted population samples based on their fluorescent intensity as indicated in Supplementary Figure1k (small Indel population= mCherry+/eGFP- , large deletion population= mCherry-/eGFP-) (page 19, line 649)

2. In terms of the inducible Cas9 cell lines, the authors state that they cannot detect Cas9 expression without Doxycycline induction, while they still observe the unwanted DSBs events in the absence of induction and claim that those are due to the leakiness of the inducible promoters. I think the authors should provide more evidence to support this conclusion, and/or they should provide a more detailed clarification of the observed results. Furthermore, the Cas9-mediated site-specific DNA cleavage is at the core of their method and the authors should show that the increase in the breaks and increase in the fraction of the mCherry+/GFP- and mCherry-/GFP- populations is Dox-dependent.

We agree with the reviewer that the Cas9-induced DNA cleavage is a core element of our method, however, the specificity of a doxycycline-inducible system is not. Our system does not rely on how tight the expression of Cas9 is regulated since we propose a two-element system taking into account the potential basal leakiness of inducible systems. In our case, if the gRNA is not transfected into the cells of interest then the induction of a DSB will not commence by any means, even if a few Cas9 molecules might exist in the cytoplasm from a not so tight regulated system.

When the reviewer suggested generating cell lines that are stably expressing gRNAs to expand the applicability for this method, we agreed that this can be a valuable addition in particular in model organisms and in hard to transfect cell lines. For this reason, we generated the cell lines with a dox inducible system integrating both the gRNA targeting eGFP and the Cas9 endonuclease. Reporting the results of these

experiments, we also raised our concerns that we are “loading” the system (Cas9 endonuclease) with “ammunition” (gRNA).

While presenting this data, our intention was to demonstrate that even though the expression of Cas9 may be hard to detect by conventional methods such as immunoblotting due to its limitations in sensitivity, this does not mean that the Cas9 is not at all produced. Indeed, here for the reviewer we present the same blot that we included in Supplementary Figure 7f, with much higher exposure, where one can start observing the Cas9 band in the uninduced samples.

Supplementary Figure 7f. Immunoblot showing inducible Cas9 expression in RPE-1 CAT-R and HEK293 CAT-R cell lines 24 h post doxycycline induction in low and high exposure.

In addition, theoretically speaking, only one molecule of Cas9 linked to a gRNA per cell can be sufficient to induce a DSB in our system and this can be measured by flow cytometry. Therefore, whether the cell produces 1 or 1.000 molecules of Cas9 is irrelevant in our system as long as the gRNA is not integrated in the cells. For this reason, we choose to work with inducible Cas9 cell lines and transfected the gRNAs to avoid potential off-target events from the Cas9 endonuclease. For this reason, while presenting this data and making the cell lines containing the integrated gRNA available for the community, we state specifically that these “all-in-one” systems should be used with caution. While in other cell lines, basal leakiness can in theory be lower since it is known that the basal expression levels can be affected by the chromosomal integration site, in the two model cell lines that we have used in this study this was not the case. Similar levels of basal leakiness in inducible CRISPR/Cas9 systems were also observed in mouse embryonic stem cells where the authors observe Indels generated by Cas9 in 33% of the samples without doxycycline induction (PMID: 25690852). Considering the high efficiency of gRNA transfection in our cell lines, and because under these conditions, DSBs can be generated without an inducible integrated gRNA, we believe that our results focusing on DNA repair choice after Cas9 induced DSBs are not influenced through gRNA delivery and we do hope that we demonstrate these points in our manuscript adequately.

In our revised manuscript, we have provided a more detailed clarification of our results, including the citation mentioned above in the discussion section and in the figure legend (Supplementary Figure 7) as stated below:

“While this system in theory can be advantageous to allow the study of DNA repair in model organisms, we also note that in such systems, unwanted

DSBs can be induced due to the basal leakiness of the inducible promoters⁶⁴. Even a few molecules of Cas9 or gRNAs that can go undetected by conventional methods (e.g. immunoblotting) can be sufficient to induce DSBs, which can be difficult to control (Supplementary Figure 7d-f) thus, these systems should be used with caution.” (page 14, line 460)

“Flow cytometry plots of engineered cell line RPE-1CAT-R stably expressing the gRNA: eGFP. Flow cytometry plots 24 and 72 h post Cas9-induction. Numbers inside plots indicate percentages of live cells. Axes show fluorescence intensities of eGFP and mCherry. Unwanted DSBs can be detected even in the absence of doxycycline induction. Please note that fluorescent proteins of the CAT-R system are also slightly expressed in the uninduced sample.” (Supplementary Figure 7e)

- 3. The authors now include estimates of the statistical significance, however, they only use them in the context of yes or no answers as to the significance of their observations. The readers might want to look at the actual values to make their own determination about relative significance of various treatments. Therefore, I think the author should make a Supplementary Dataset with the actual p-values and %-ages for every sorted fraction for every drug treatment and gene knockout (eg, ATM, PARP etc) tested in this work.**

Following the reviewer’s suggestion, we included Source Data files for (1) KO screen including p-value calculations, and (2) Drug screens, which include

- *the KO raw datasets including p-value calculations*
- *all the drug screen raw datasets.*
- *values of the CRISPR/Cas9 arrayed screen*

- 4. The authors provide 3 supplementary tables, but they only mention one of them in the text.**

Following the reviewer’s suggestion, we annotate the supplementary tables in the text as follows:

- *The supplementary Table2 that presents the goodness of fit statistics for the drug screen is referenced at (page 7, line 215). Other tables are now included in the Source Data file.*

REVIEWERS' COMMENTS:

Reviewer #2 (Remarks to the Author):

The authors have overall addressed my queries.